# A novel ferroptosis-related signature for predicting prognosis, immune characteristics, and treatment prediction in hepatocellular carcinoma

Chengting Wu[1☉], Xinyuan Chen[1☉], Yu Zhang[1], Yuanqin Du[1], Jian Xu[1], Yujiao Peng[1], Lu Lu[1], Jingjing Huang [2,3]*, Hongna Huang[2]*

**1** Guangxi University of Chinese Medicine, Nanning, China, **2** The First Affiliated Hospital of Guangxi University of Chinese Medicine, Nanning, China, **3** Guangxi Key Laboratory of Translational Medicine for Treating High-Incidence Infectious Diseases with Integrative Medicine, Nanning, China,

☉ These authors have contributed equally to this work.
* 55869563@qq.com (JH); 18776985262@163.com (HH)

## Abstract

Hepatocellular carcinoma (HCC) is a malignant tumor with a high incidence and fatality. The occurrence and progression of HCC are tightly linked to ferroptosis, a unique type of cell death. To accurately predict the prognosis, immunological traits, treatment sensitivity, and drug prediction for patients with HCC, this work attempts to develop a novel ferroptosis-related gene signature (nFRGs). Several machine learning techniques were applied to build the nFRGs model utilizing data from The Cancer Genome Atlas (TCGA) and GSE14520 datasets. Different analysis packages in R version 4.4.1 were also used for prognosis analysis, molecular function analysis, somatic mutation analysis, immunotherapy response analysis, immunotherapy evaluation, drug sensitivity analysis, and drug prediction to compare the differences between the low-risk and high-risk groups. The nFRGs model includes five ferroptosis-related genes (KIF20A, NT5DC2, G6PD, SLC7A11, and EZH2). The results indicate that nFRGs are an independent prognostic risk factor for HCC patients, and patients in the high-risk group have a worse prognosis. Our nFRGs model shows better accuracy and reliability in predicting the prognosis of HCC patients than other existing ferroptosis-related gene models. Both the high- and low-risk groups of nFRGs had differentially expressed genes (DEGs) enriched in pathways mostly associated with immunological traits and tumor progression. The high-risk group exhibited clear immune escape characteristics, with significant upregulation in the expression of immune checkpoints and TIDE scores. Furthermore, IPS analysis also revealed that the high-risk group is less responsive to immunotherapy, while the low-risk group showed a better potential for immune therapy response, which further highlights the potential of nFRGs as a predictor for immunotherapy outcomes. This suggests a stronger immune suppression status in high-risk patients,

which permits unrestricted use, distribution, and reproduction in any medium, provided the original author and source are credited.

**Data availability statement:** All relevant data are within the manuscript and its Supporting Information files.

**Funding:** This research was funded by the National Natural Science Foundation of China grant number [No. 82460957], the Guangxi Natural Science Foundation Project [No. 2022GXNSFAA035460, No. 2024GXNSFDA010005], and the Guangxi Graduate Education Innovation Program [No. YCSW2023395]. The funders had no role in study design, data collection and analysis, decision to publish, or preparation of the manuscript.

**Competing interests:** The authors have declared that no competing interests exist.

**Abbreviation:** HCC, Hepatocellular carcinoma; nFRGs, novel ferroptosis-related gene signature; TCGA, The Cancer Genome Atlas; DEGs, Differentially expressed genes; ICIs, immune checkpoint inhibitors; TIDE, Tumor Immune Dysfunction and Exclusion; FRGs, Ferroptosis-related genes; OS, Overall survival; GEO, Gene Expression Omnibus; Cox, Cox proportional hazards regression model; C-index, Concordance Index; AUC, Area Under the Curve; HPA, Human Protein Atlas; ROC, Receiver operating characteristic; HR, Hazard ratio; PCA, Principal Component Analysis; GO, Gene Ontology; KEGG, Kyoto Encyclopedia of Genes and Genomes; GSEA, Gene Set Enrichment Analysis; MsigDB, Molecular Signatures Database; STRING, Search Tool for the Retrieval of Interacting Genes; TME, Tumor immune microenvironment; IC50, Half maximal inhibitory concentration; DsigDB, Database of Signatures of Gene Expression; Coef, Coefficients; SNPs, Single nucleotide polymorphisms

potentially leading to a poorer response to immune checkpoint inhibitors (ICIs). In contrast, the low-risk group displayed lower immune escape features, making them potentially more susceptible to immune responses. Additionally, there were significant differences in gene mutations, molecular functions, and other factors between the low-risk and high-risk groups. Lastly, our investigation predicted possible medications that would work well for the model and found sensitive chemotherapeutic and targeted medications for both high-risk and low-risk groups. In conclusion, nFRGs could serve as a novel prognostic biomarker, providing valuable insights for personalized treatment strategies.

## Introduction

The incidence and mortality rates of HCC, one of the most prevalent malignant tumors in the world, are increasing yearly [1,2]. The prognosis of HCC is still not adequate despite major advancements in treatment approaches, such as immunotherapy and targeted therapy [3–6]. Therefore, increasing the survival rate and quality of life for patients with HCC requires the development of novel predictive models and treatment approaches.

Ferroptosis, a form of programmed cell death induced by iron-dependent lipid peroxidation, plays an important role in the occurrence and progression of various tumors [7,8]. Ferroptosis has been implicated in the development, progression, and mechanisms of treatment resistance of HCC in recent years [9,10]. By controlling lipid peroxidation and iron homeostasis inside the tumor cells, ferroptosis influences the survival and death of tumor cells [11]. Ferroptosis-related genes (FRGs) have shown great promise in the prognostic assessment of HCC in a growing number of studies [12–17]. As a result, prognostic models built using these genes can enhance HCC prediction skills while also providing a scientific foundation for individualized care.

HCC immunological milieu is essential to the development and spread of the tumor. The immunological microenvironment of HCC affects immune cell invasion, immune escape, and immune tolerance [18,19]. Ferroptosis's function in controlling the immunological milieu of HCC has drawn more and more attention. According to studies, ferroptosis influences immune cell activity in the tumor immunological milieu, which modifies the immune response in addition to directly affecting the survival of HCC cells [20,21]. To comprehend the immune escape mechanisms of HCC and create novel immunotherapy approaches, it is crucial to investigate the relationships between ferroptosis-related genes and the immunological milieu of HCC.

Furthermore, with a deeper understanding of the mechanisms of ferroptosis, an increasing number of drugs have been found to have the potential to regulate ferroptosis. By focusing on iron metabolism or lipid peroxidation pathways, these medications can cause ferroptosis in HCC cells. They can also improve the efficacy of immunotherapy by altering the immune microenvironment [22,23]. Thus, ferroptosis-related gene-based drug prediction models offer fresh perspectives on the accurate

management of HCC and could potentially be a useful strategy for enhancing the therapeutic result of HCC, especially when combined with immunotherapy and targeted medications.

In conclusion, new routes for the clinical diagnosis and treatment of HCC are opened by the potential of FRGs in prognostic assessment, immunological microenvironments, and medication treatment. Ferroptosis-related gene-based prognostic models, immune microenvironment analysis, and medication prediction will all contribute to better treatment outcomes and a higher standard of living for HCC patients.

## Materials and methods

HCC patients' RNA sequencing and related clinical data were sourced from the TCGA database (https://portal.gdc.cancer.gov/). The TCGA-LIHC dataset includes 50 normal tissue samples and 369 HCC samples for differential expression analysis. After excluding samples with overall survival (OS) < 30 days or incomplete clinical information, 341 HCC samples were included in the subsequent analysis (Table 1). Among the 341 HCC patients in the TCGA cohort, the OS distribution was as follows: 219 patients survived, and 122 patients died, with a median OS time of 630 days.The Gene Expression Omnibus (GEO) database (GSE14520) includes 241 normal tissue samples and 247 HCC samples for differential expression analysis. After excluding samples with overall survival (OS) < 30 days or incomplete clinical information, 221 HCC samples were included in the subsequent analysis (Table 1). Among the 221 HCC patients in the GSE14520 cohort, the OS distribution was as follows: 136 patients survived, and 85 patients died, with a median OS time of 1569 days.To identify ferroptosis-related genes (FRGs) associated with the prognosis of HCC, we gathered FRGs from the FerrDb database (http://www.zhounan.org/ferrdb/), which includes genes that promote, inhibit, or are involved in ferroptosis. The research workflow is displayed in Fig.1.

### DEGs analysis

Using the "DESeq2" R program, the TCGA database's DEGs between tumor and normal samples were filtered. Either "Wald" or "LRT", which will then use either Wald significance tests (defined by nbinomWaldTest), or the likelihood ratio test on the difference in deviance between a full and reduced model formula (defined by nbinomLRT).The filtering limits were $|log2(FC)| > 1$ and false discovery rate (FDR) < 0.05.

**Table 1. Baseline Characteristics Comparison of HCC Patients from the GSE14520 and TCGA-LIHC Databases.**

|  | [ALL] | GSE14520 | TCGA-LIHC | p.overall |
|---|---|---|---|---|
|  | N = 562 | N = 221 | N = 341 |  |
| OS: |  |  |  | 0.579 |
| Alive | 355 (63.2%) | 136 (61.5%) | 219 (64.2%) |  |
| Dead | 207 (36.8%) | 85 (38.5%) | 122 (35.8%) |  |
| OS.time | 1000 (713) | 1213 (647) | 863 (720) | <0.001 |
| Gender: |  |  |  | <0.001 |
| Female | 139 (24.7%) | 30 (13.6%) | 109 (32.0%) |  |
| Male | 423 (75.3%) | 191 (86.4%) | 232 (68.0%) |  |
| Age | 56.0 (12.9) | 50.8 (10.6) | 59.3 (13.2) | <0.001 |
| TNM.staging: |  |  |  | 0.002 |
| Stage I | 276 (49.1%) | 93 (42.1%) | 183 (53.7%) |  |
| Stage II | 156 (27.8%) | 79 (35.7%) | 77 (22.6%) |  |
| Stage III | 127 (22.6%) | 49 (22.2%) | 78 (22.9%) |  |
| Stage IV | 3 (0.53%) | 0 (0.00%) | 3 (0.88%) |  |

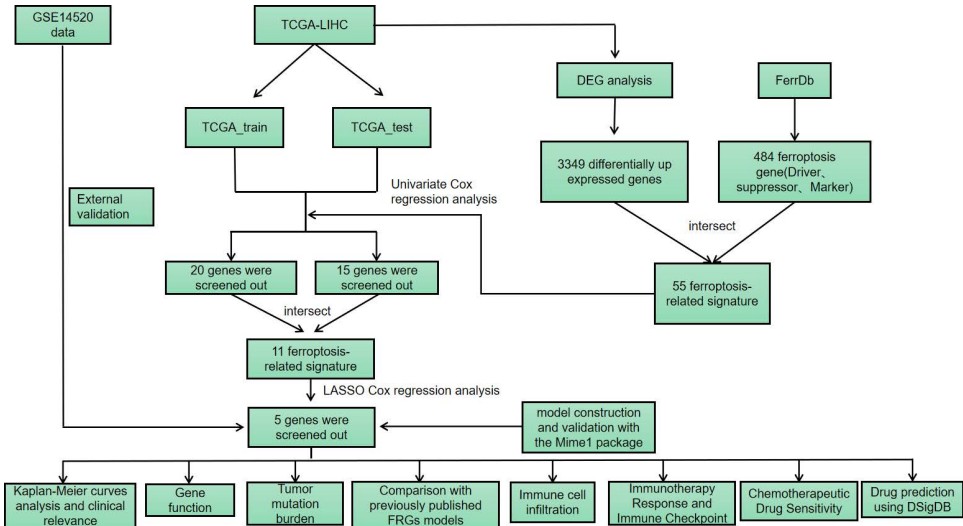

**Fig 1. The workflow of this study involves the use of TCGA-LIHC (The Cancer Genome Atlas Liver Hepatocellular Carcinoma dataset), GSE14520 (GEO dataset), FerrDb (Ferroptosis Database), DEG (Differentially Expressed Genes), and LASSO (Least Absolute Shrinkage and Selection Operator).**

## Construction and analysis of nFRGs

The tumor samples were randomly split into training and testing sets at an 8:2 ratio using the "caret" R package. To find FRGs with prognostic significance, we first chose the intersection of FRGs and upregulated DEGs. Subsequently, we performed univariate Cox proportional hazards regression model (Cox)regression analysis independently in both the training and testing sets.

We employed the "Mime1" R package, which offers a combined analysis of ten popular machine learning techniques to produce different prediction models, to choose the best model [24]. To select the best-performing model, we ranked the models according to their performance using Mime1 and computed the Concordance Index (C-index) and the 1-, 2-, and 3-year Area Under the Curve (AUC)values for each model in the training and validation sets. After comparing the C-index and AUC values of various algorithm combinations, we selected the model that performed well in both the training and validation sets. We then used this model for further analysis and prediction. Along with RNA-level analysis, we also evaluated the protein expression of prognostic genes in normal liver and HCC tissues using immunohistochemistry staining maps that we acquired from the Human Protein Atlas (HPA) database (https://www.proteinatlas.org/).

Next, we used the "survminer" and "survival" packages for analysis and divided the training set into high-risk and low-risk groups based on the median risk score of the model. The "survminer" package was used to perform Kaplan-Meier survival curve analysis, and time-dependent receiver operating characteristic (ROC) curves and AUC values for 1-, 2-, and 3-year survival were used to evaluate the model's accuracy. To evaluate the target factors' influence on prognosis, we also conducted univariate Cox regression analysis on datasets like TCGA_train, TCGA_test, TCGA, and GSE14520. Each feature's hazard ratio (HR) and 95% CI were determined as part of the study, and the statistical significance (P-value) of each was assessed. The results from various datasets were then combined using meta-analysis, which used fixed-effects and random-effects models to assess the robustness and consistency of characteristics across datasets. R software was used for all statistical analyses, and $p < 0.05$ was chosen as the significance level.

Finally, using the median risk score from the TCGA and GSE14520 datasets to categorize the samples into high-risk and low-risk groups, we analyzed the distribution of risk scores, patient survival status, and gene expression. We then

performed Principal Component Analysis (PCA) on the two risk groups to assess the differences in their multidimensional distributions. To evaluate the performance of our model, we compared it with previous ferroptosis-related prognostic gene models, using HR, C-index, and AUC.

## Nomogram construction and validation

To ascertain whether the developed risk model could function as an independent prognostic factor for HCC, univariate and multivariate Cox regression analyses were conducted on the clinical features and risk groups of the TCGA and GSE14520 datasets using the "survival" package. Lastly, a nomogram was created using the "rms" R package to forecast the OS of HCC patients at one, two, and three years.

## Functional enrichment analysis

We used the "limma" R package to perform DEGs on the two subgroups to examine the biological function and molecular mechanisms that differ between the high-risk and low-risk groups according to the median risk score of nFRGs. Next, we used the "clusterProfiler" program to do enrichment analysis on the derived DEGs using Gene Ontology (GO) and the Kyoto Encyclopedia of Genes and Genomes (KEGG). To identify possible biological pathways and functions, we lastly carried out Gene Set Enrichment Analysis (GSEA) using the "clusterProfiler" and "enrichplot" packages based on reference gene sets from the Molecular Signatures Database (MsigDB) database (symbols. gmt v7.0).

## Protein-Protein Interaction (PPI) network construction

First, we used Pearson correlation analysis to assess the relationship between predictive genes. The "limma" packages were then used to conduct DEGs comparing high-risk and low-risk groups. These genes were added to the Search Tool for the Retrieval of Interacting Genes (STRING) online database (https://cn.string-db.org/) to create their PPI network, with a 0.4 confidence level. Cytoscape was used to visualize the PPI network and show the interaction network of the chosen nFRGs.

## Gene mutation analysis

We obtained genetic variation data from the TCGA dataset and used the mutation data to determine each patient's TMB score. Next, we examined the association between TMB and survival rate and contrasted the TMB variations between the high-risk and low-risk groups. We examined the quantity and kind of gene mutations in the high-risk and low-risk groups in the TCGA and GSE14520 datasets using the "Maftools" packages. We determined the main types of mutations and evaluated the prognostic-related gene mutation status in HCC. Furthermore, we examined the relationship between the main nFRG mutation types and the levels of mRNA expression in the TCGA and GSE14520 datasets.

## Immune characteristics

We compared the immune cell infiltration disparities between high-risk and low-risk groups, evaluated the relationship between risk scores and 22 immune cell types, and quantified immune cell infiltration and immunological function for each sample using the "GSVA" package and the "CIBERSORT" algorithm. The immunological properties of the tumor immune microenvironment (TME) were assessed using the "estimate" package. The ESTIMATE score is a method that calculates the immune and stromal scores of the tumor microenvironment. These scores are indicative of tumor purity and the extent of immune infiltration, which can influence prognosis and treatment responses in HCC.Additionally, the expression variations of common immunological checkpoints in each subgroup were examined using the "limma" c. To predict how HCC patients will react to immunotherapy, the TIDE algorithm was used to determine the TIDE score for each sample and examine the variations in TIDE scores between high-risk and low-risk groups.The TIDE algorithm was employed to predict potential responses to immune checkpoint blockade in HCC, where a lower TIDE score indicates a better response to immunotherapy.The TIDE data used in this study was obtained from the online database (http://tide.dfci.harvard.edu/faq/).

## Drug sensitivity analysis

We employed the "oncoPredict" R packages to evaluate the medication sensitivity of nFRGs in both high-risk and low-risk groups of HCC patients. Intending to offer recommendations for individualized treatment, this software forecasts medication sensitivity using gene expression data. We determined the expected half maximal inhibitory concentration (IC50) values for each of the 198 medications we chose for the analysis in patients with HCC. The IC50 value, which indicates how sensitive cells are to a drug, is the concentration of the drug that suppresses cell growth by 50%. We were able to determine viable treatment choices with varying drug sensitivities in various risk populations by comparing the IC50 values between the high-risk and low-risk groups.

## Drug prediction based on nFRGs

We used the Database of Signatures of Gene Expression (DsigDB) database to further investigate possible therapeutic medications associated with the nFRGs in the study. We examined medications linked to the target genes based on their expression patterns. It is well known that the chosen medications significantly regulate particular genes.

## Statistical analysis

R software (version 4.4.1) and the online analysis tool MicrobeX (https://www.bioinformatics.com.cn/) were used for all statistical analyses. With a significance level of $p < 0.05$, this study used several R packages, including Mime1, limma, survival, and caret. The criteria for statistical significance were $p < 0.05$, $p < 0.01$ and $p < 0.001$.

## Results

### nFRG construction and validation

50 normal samples and 369 HCC samples were among the 419 samples that were obtained from the TCGA database. 3349 elevated DEGs between tumor and normal tissues were found using differential analysis (Fig 2A). Furthermore, the FerrDb database yielded 484 FRGs. 55 differentially expressed FRGs were chosen after calculating the overlap of DEGs and FRGs (Fig 2B and S1 Table).

The 55 differentially expressed FRGs were subjected to univariate Cox regression analysis in the training and testing set that were separated from the TCGA dataset. According to the findings, 15 genes were linked to prognosis in the testing set and 20 genes were linked to prognosis in the training set. Eleven genes that were prognostic in both datasets were found by intersecting these two gene sets (Fig 2C). The number of feature genes was then decreased using LASSO regression analysis. Following the preprocessing of the gene expression data, a LASSO regression model was built. To maximize the predictive performance and control complexity of the model, the ideal λ value was chosen by cross-validation. Five nFRGs were ultimately chosen (Figs 2D and 2E): G6PD, SLC7A11, KIF20A, NT5DC2, and EZH2.

It should be mentioned that only 10 genes were shown in the Coefficients (Coef) of the LASSO regression model, even though 11 genes were found in the intersection analysis. This is probably because several genes' regression coefficients were compressed to zero, meaning that they were not included in the current analysis since they lacked the necessary predictive power. Finally, we used a heatmap to show the expression of these 5 genes in the TCGA dataset (Fig 2F), and we examined the immunohistochemical expression of 4 of these genes in normal and HCC tissues using data from the HPA database. The findings demonstrated that while G6PD, KIF20A, NT5DC2, and EZH2 were not found in normal tissues, they were expressed in HCC tissues to varied degrees. Interestingly, the HPA database did not contain any immunohistochemistry expression data for SLC7A11 in liver tissue (S1 Fig).

Using the "Mime1" machine learning package, we fitted 87 predictive models to the TCGA-train dataset and determined the C-index for each model in the training and validation datasets (Fig 3A). The results show that the StepCox[forward]

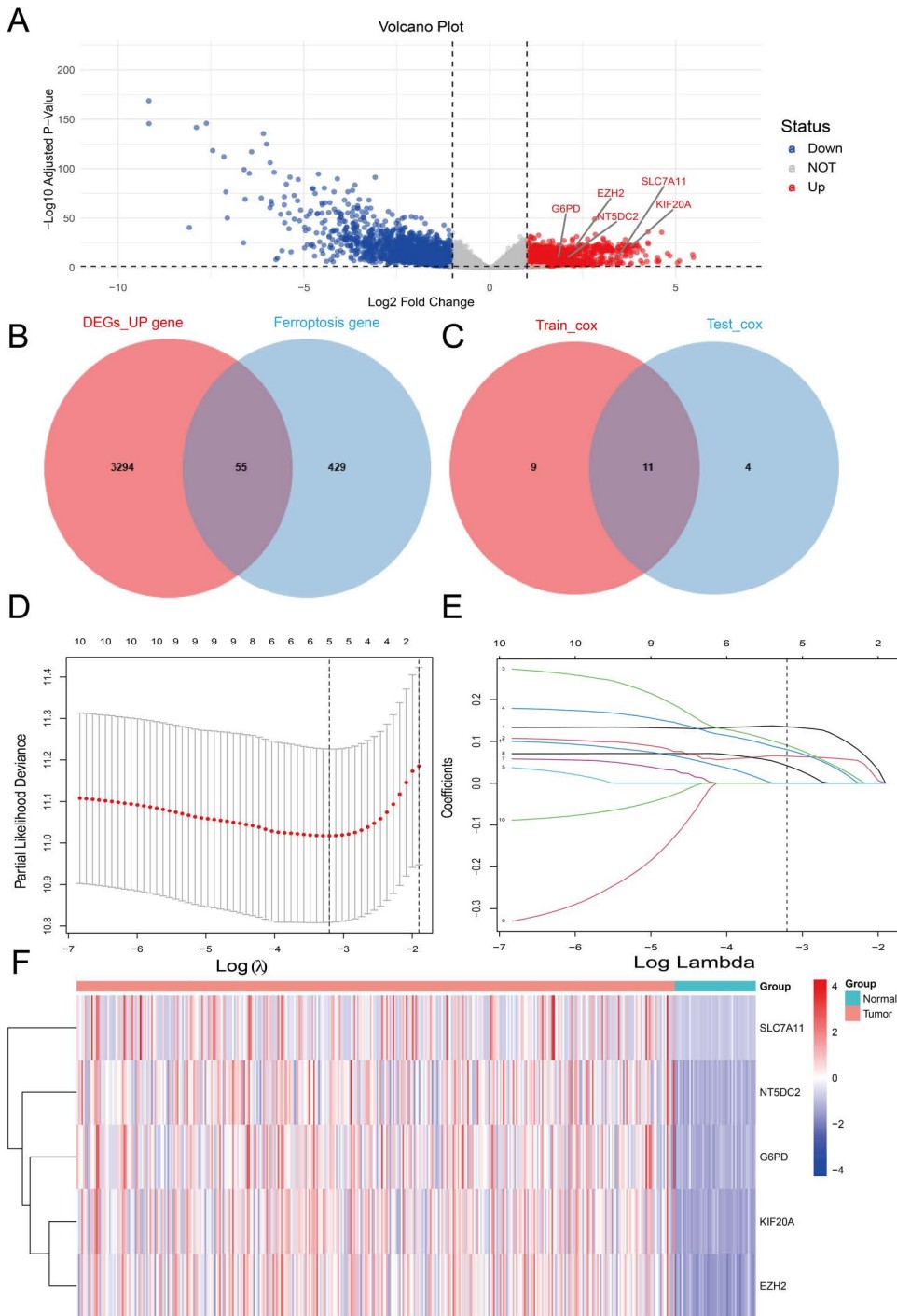

**Fig 2. Differential gene analysis results (A): Volcano plot showing the expression of differential genes in normal and tumor samples from the TCGA dataset.** (B) Venn diagram of upregulated differentially expressed genes and ferroptosis-related genes (FRGs). (C) Venn diagram of univariate Cox genes in TCGA_train and TCGA_test datasets. (D) Lasso regression coefficient plot of the 11 FRGs. (E) Tuning parameters in the Lasso model. (F) Heatmap of the expression of five gene features in normal and tumor tissues from the TCGA dataset.

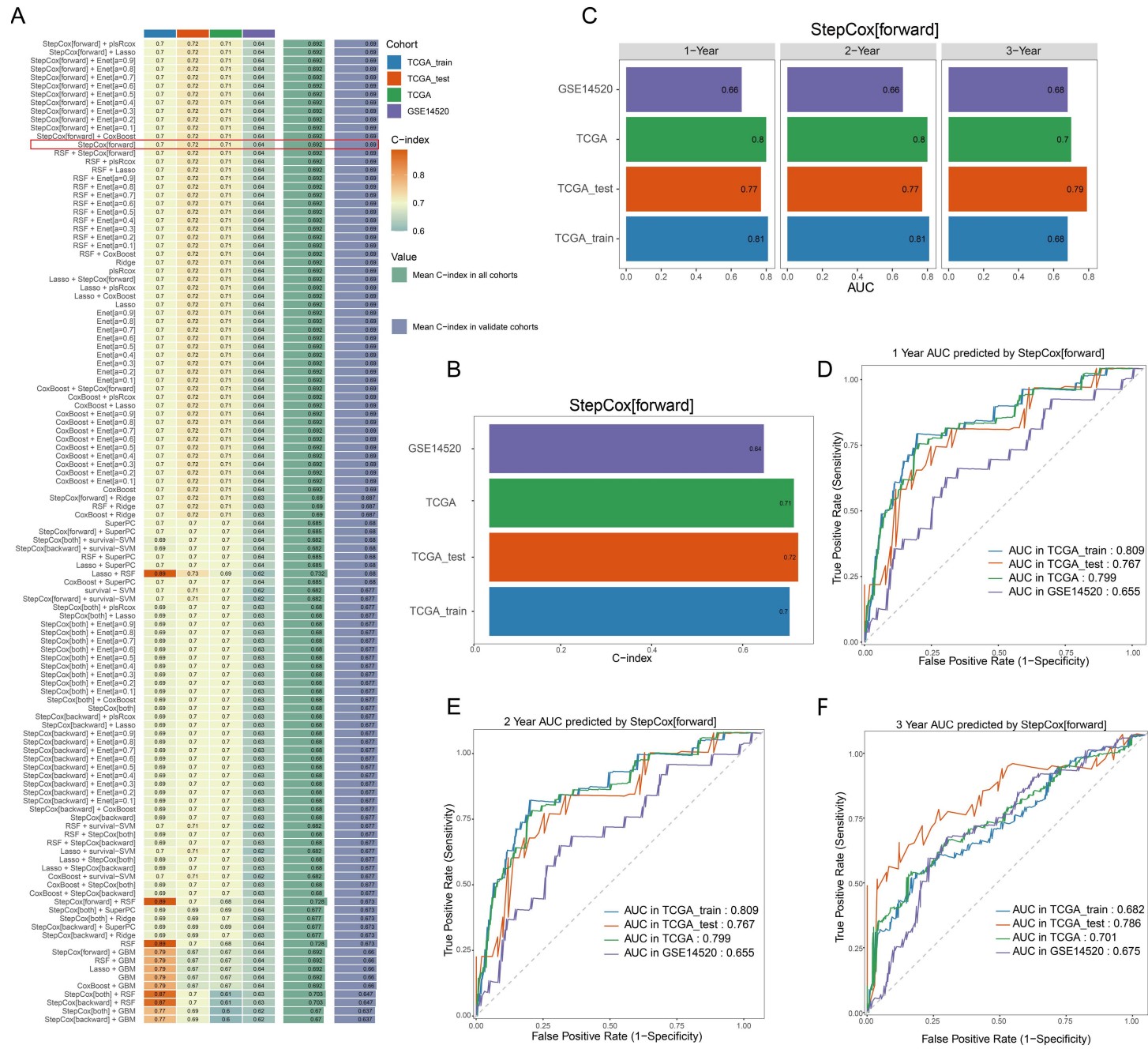

**Fig 3. Model construction and validation (A) Construction of the model based on multiple machine learning algorithms.** (B) C-index of the StepCox[forward] model across different datasets. (C) AUC values of the StepCox[forward] model for 1-year, 2-year, and 3-year outcomes in different datasets. (D-F) ROC curves for 1-year, 2-year, and 3-year outcomes in various datasets.

model performed best with a simple model in both the training and validation datasets, with average C-index values of 0.692 and 0.69, respectively (Fig 3A). In the training and validation datasets, its C-index values were 0.7, 0.72, 0.71, and 0.64, in that order (Fig 3B). The StepCox[forward] model performed well in the 3-year AUC values, according to the additional computation of the AUC values for each model at 1, 2, and 3 years (S2 Fig).

Therefore, StepCox[forward] was selected as the best model, and its AUC value was extracted (Fig 3C). To assess the StepCox[forward] model's prognostic prediction performance, we then plotted the 1-, 2-, and 3-year ROC curves for the training and validation datasets (Figs 3D-F).

Finally, based on the median of the risk scores calculated by the StepCox[forward] model in the "Mime1" package, patients in both the training and validation cohorts were divided into high-risk and low-risk groups. The following formula is used to determine the risk score: $0.071 \times$ KIF20A expression $+ 0.1015 \times$ SLC7A11 expression $+ 0.1239 \times$ NT5DC2 expression $+ (0.1452 \times$ G6PD expression$) + (0.1514 \times$ EZH2 expression$)$. High-risk patients in the training sample are substantially linked to a worse OS, according to the Kaplan-Meier survival curve (Fig 4A). Kaplan-Meier survival curves were analyzed using the log-rank test to compare survival differences between high- and low-risk groups.The validation set showed similar results (Figs 4B–D).To further validate whether the difference in diagnosis year affects survival analysis results, we divided the patients into two groups based on their diagnosis year: pre-2007 and post-2007, with 2007 marking a significant change in the treatment era. Initial survival analysis showed that diagnosis year alone was associated with differences in overall survival. However, when patients were stratified based on their risk scores, the survival differences between the pre-2007 and post-2007 groups were no longer statistically significant. This suggests that diagnosis year is not a decisive factor influencing survival outcomes. Instead, survival differences appear to be more strongly driven by the risk group (high-risk vs low-risk), which further supports the stability and robustness of the constructed risk model across different treatment eras (S3 Fig and S2 Table).

To further evaluate the survival prognostic value of the selected features across different datasets, we performed univariate Cox regression analysis in each dataset and used meta-analysis to integrate the results. Univariate Cox regression was performed to estimate the hazard ratios (HR) for each feature's impact on OS. According to specific findings, the chosen feature's HR in the training set was 1.76, with a 95% CI of 1.16–2.67 and a P-value of 0.008, suggesting a significant correlation with OS (Fig 4E). The P-values were less than 0.05 in every test set, indicating statistical significance.

In the meta-analysis, we used both a random-effects model and a fixed-effects model to integrate the results from different datasets. The random-effects model was employed to account for heterogeneity across datasets, while the fixed-effects model was used when assuming homogeneity across the studies.With a P-value of 0.001 and an HR of 2.08 (95% CI: 1.66–2.61) after merging, the random-effects model showed that this trait had a consistent and statistically significant impact on survival prognosis across several datasets. When combined, the meta-analysis provided additional confirmation of this feature's resilience and prognostic predictive significance across other datasets.

## Risk Model Evaluation

To better stratify risk scores into high-risk and low-risk groups, we compared the use of the first quartile, median, and third quartile in univariate Cox regression analysis of clinical prognosis to select the optimal grouping method. The results showed that in the TCGA dataset, the low-high risk groups defined by the third quartile exhibited the best effect and significance, while the median grouping also showed significance, whereas the first quartile grouping did not demonstrate significant effects. In the GSE14520 dataset, only the median grouping showed significance, while both the first quartile and third quartile groupings did not show significant effects. Therefore, we concluded that using the median to define low and high-risk groups has stronger generalizability and can be applied in the GSE14520 dataset. Based on this, we divided patients in both the TCGA and GSE14520 cohorts into high-risk and low-risk groups according to the median risk score(S4 Fig).High-risk and low-risk patients were significantly clustered in both cohorts in the PCA analysis (Figs 5A and 5E). Gene expression levels, survival time, survival status, and distribution of risk scores were also examined. The TCGA cohort's findings (Figs 5B–D) demonstrated that five genes had higher expression levels and that high-risk individuals died at higher rates. A similar pattern was also seen in the GSE14520 cohort's data (Figs 5F-H). Overall, these results indicate that our prognostic model demonstrates accuracy and robustness across different datasets.

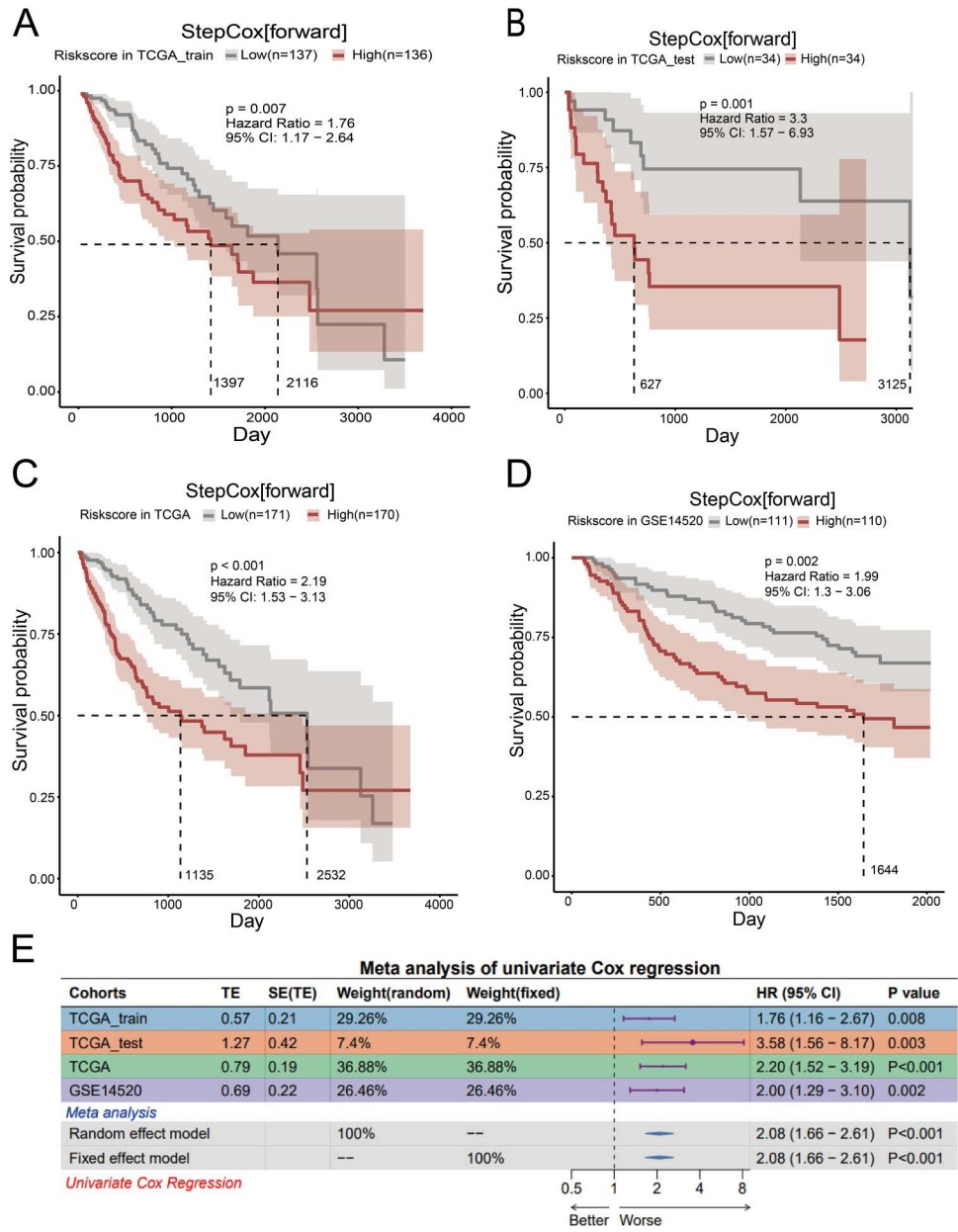

**Fig 4. Validation of the prognostic model (A-D)** Kaplan-Meier survival curves for the overall survival of two risk subgroups in TCGA_train, TCGA_test, TCGA, and GSE14520. Dashed line represents the Median Overall Survival (OS). (E) Meta-analysis of univariate Cox regression. High: High risk group based on the ferroptosis-related gene signature (nFRGs). Low: Low risk group based on the ferroptosis-related gene signature (nFRGs).

## Comparison with previously published ferroptosis-related gene prognostic models

Through a literature search, we collected 6 ferroptosis-related gene prognostic models published in recent years and organized the corresponding genes, Coef, and PMID numbers for each model (Table 2 and S3 Table). This collection offers background knowledge and references for assessing our model, enabling a thorough comparison with earlier versions.

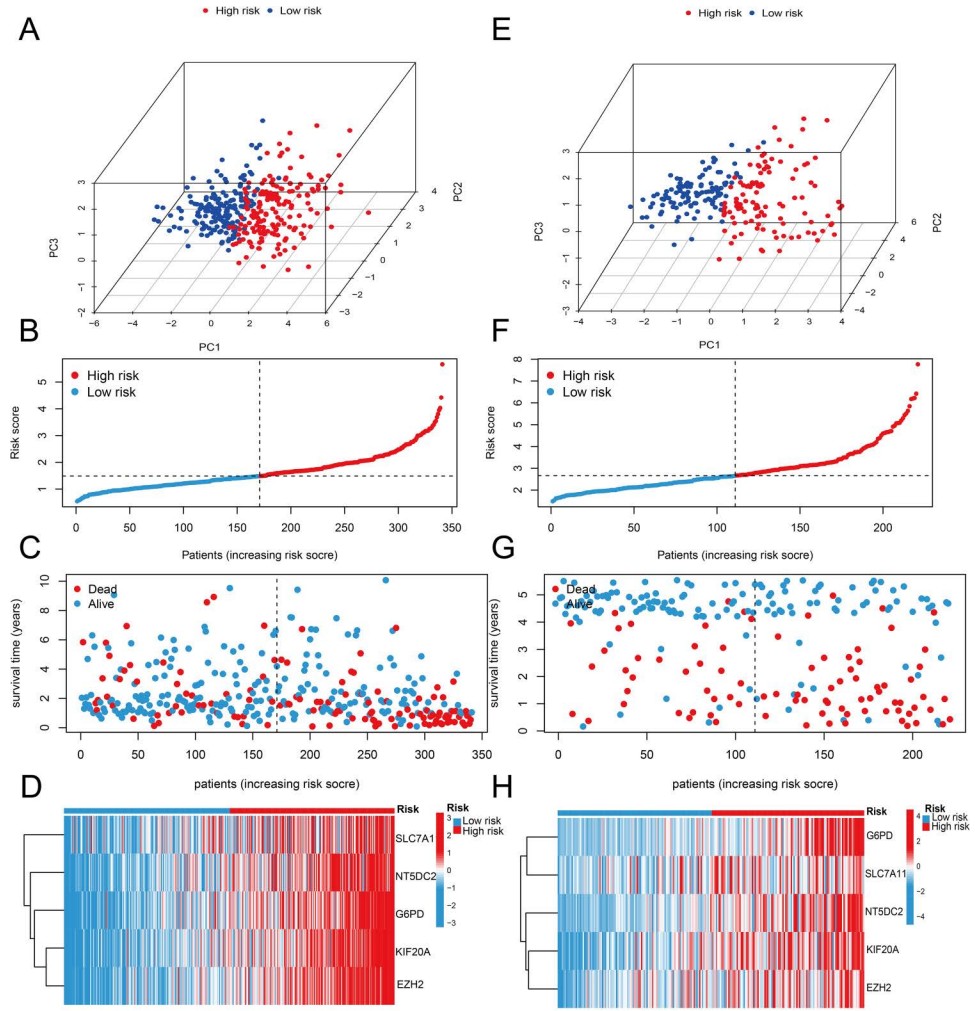

**Fig 5. Performance of the prognostic model risk subgroups in the TCGA and GSE14520 datasets.** (A-D) The PCA 3D analysis, distribution of risk scores, correlation between survival time and survival status for each patient, and heatmap of the prognostic model genes in the risk subgroups for TCGA. (E-H) The PCA 3D analysis, distribution of risk scores, correlation between survival time and survival status for each patient, and heatmap of the prognostic model genes in the risk subgroups for GSE14520.

**Table 2. Comparison between our ferroptosis-related gene prognostic model and previously published models.**

| Model | Gene List | References (PMID) |
|---|---|---|
| Our model | G6PD,KIF20A,EZH2,NT5DC2,SLC7A11 | – |
| Signature1 | AURKA, CDCA3, STMN1, SLC7A11, G6PD, NT5DC2, NQO1 | 36290827 [12] |
| Signature2 | AKR1C3, ATIC, G6PD, GMPS, GNPDA1, IMPDH1, PRIM1, RRM2, TXNRD1 | 34377010 [13] |
| Signature3 | SLC7A11, SLC1A5, TFRC, CARS1, RPL8 | 35847985 [14] |
| Signature4 | SLC2A1, SRXN1, FLT3, UBC, RRM2, HILPDA, KEAP1, ACSL3, PRDX6, ZFP69B, AKR1C3 | 36647389 [15] |
| Signature5 | HRAS, SLC2A1, NRAS, MAPK3, RRM2 | 35938001 [16] |
| Signature6 | G6PD, NRAS, HRAS, TIMM9, MYCN | 37143953 [17] |

The findings demonstrate that our model performs at an intermediate to above-average level in both the average C-index and the C-index of the validation cohort when compared to existing prognostic models (Fig 6A). The AUC values for 1-, 2-, and 3-year survival showed a similar pattern (S5 Fig). We contrasted our model with six other published HCC FGRs prognostic models (Signature1 through Signature6 and Table 1) to confirm its prognostic prediction power. HR are represented by different colors in the heatmap shown in Fig 6B, where higher HR is indicated by darker hues. Statistical significance is indicated by the following symbols: ** for $p < 0.01$, and * for $p < 0.05$.

Our model showed a strong prognostic prediction capacity across four datasets when comparing the training and testing cohorts. The HR is near 4, with significance at $p < 0.01$, especially in the TCGA_test cohort, suggesting a better capacity to differentiate between individuals at high and low risk. The HR values of the other five published models, on

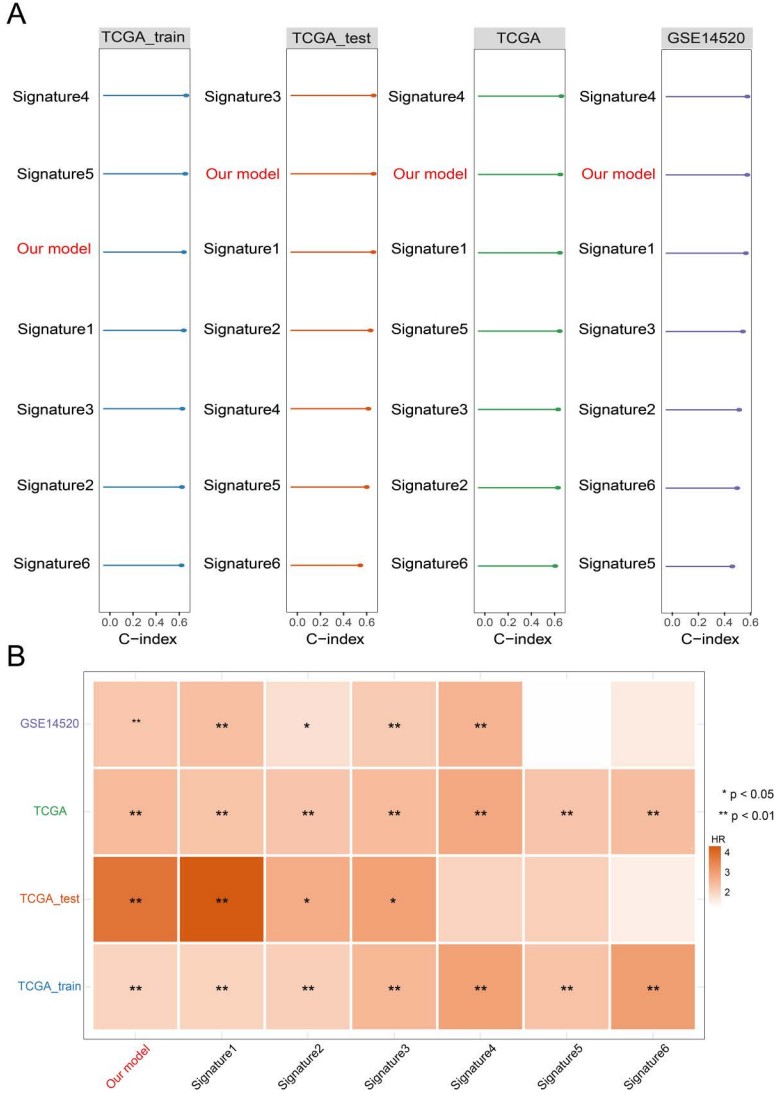

**Fig 6. Comparison with previously published ferroptosis-related gene models.** (A) C-index for each model in the different datasets. (B) HR heatmap for each model in the different datasets, with different colors representing different hazard ratios (HR), where darker colors indicate higher HR. Statistical significance is indicated by symbols: * for $p < 0.05$ and ** for $p < 0.01$, indicating statistical significance.

the other hand, were comparatively weak and their significance was not as strong as that of our model, even though all models performed well in the training set ($p < 0.01$), except the Signature1 model, in the testing set. This implies that when extrapolated to separate datasets, their prognostic efficacy is lower, most likely as a result of variations in data processing techniques that result in poorer prediction performance. The robustness and superiority of this model in prognostic prediction for HCC are further validated by the comparison findings of our model with prior models across four datasets, which show higher C-index, AUC, and HR values.

## Nomogram construction and validation

To find out if risk score grouping and other clinical characteristics could be independent variables for HCC prognosis, we employed univariate and multivariate Cox regression analysis. The TCGA clinical characteristics dataset'sunivariate Cox analysis revealed that risk score grouping, T staging, and TNM staging were all substantially correlated with prognosis ($p < 0.05$) (Fig 7A). Only the risk score grouping displayed comparable significant results in the multivariate Cox analysis (Fig 7B). These findings imply that the risk score may function as a stand-alone predictor of HCC.

We created a nomogram based on the risk score using the findings of the univariate and multivariate analyses from the TCGA entire cohort (Fig 7C). The nomogram's ability to accurately predict the prognosis of patients with HCC was validated.

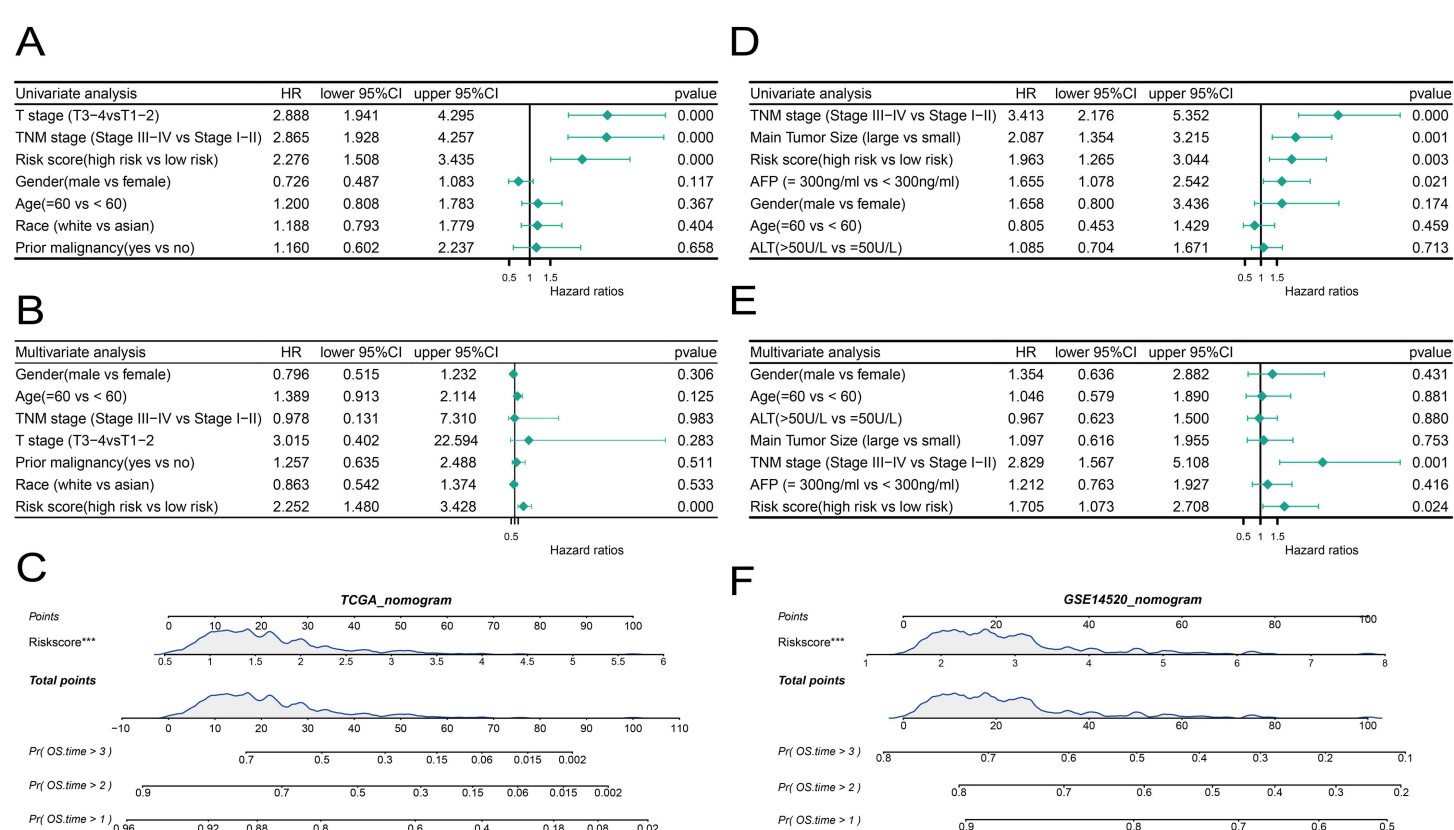

**Fig 7. Construction and validation of the nomogram (A-C) Forest plots of univariate and multivariate Cox analyses for risk score subgroups and clinical features in the TCGA dataset, along with the construction of the risk score nomogram.** (D-F) Forest plots of univariate and multivariate Cox analyses for risk score subgroups and clinical features in the GSE14520 dataset, along with the construction of the risk score nomogram.

We also performed the same analysis on the GSE14520 clinical characteristics dataset's external validation cohort. TNM staging, alpha-fetoprotein levels, tumor size, and risk score grouping were all substantially correlated with prognosis ($p < 0.05$), according to the univariate Cox analysis of the GSE14520 dataset (Fig 7D). Risk score grouping and TNM staging both produced significant results in the multivariate Cox analysis (Fig 7E). The risk score's function as an independent prognostic factor for HCC is further supported by these findings.

Based on the univariate and multivariate analysis results from GSE14520, we constructed a nomogram using only the risk score from GSE14520 to compare with the TCGA nomogram (Fig 7F). These findings confirm once more how well the nomogram predicts the prognosis of patients with HCC.

## DEGs functional enrichment analysis for high vs. low-risk groups

These DEGs were mainly involved in biological processes related to chromosome separation, nuclear chromosome separation, sister chromatid separation, and mitotic chromosome separation, according to the GO analysis's GO-biological process (GO-BP) enrichment results. The DEGs were substantially enriched in cellular components such as centromere regions, condensed chromosomes, and chromosome regions, according to the GO-cellular component (GO-CC) enrichment data. These genes were linked to molecular functions such as arachidonic acid monooxygenase activity, steroid hydroxylase activity, and microtubule binding in the GO-molecular function (GO-MF) study (Figs 8A and 8B).

According to the KEGG pathway enrichment analysis, these genes were found to be enriched in a number of biological pathways, including those pertaining to complement and coagulation cascades, the cell cycle, carbon metabolism, drug metabolism, DNA replication, and chemical carcinogenesis-DNA adducts (Figs 8C and 8D).

In the GSEA analysis, the results of the C7 gene set (Fig 8E) suggested that the immune microenvironment of the high-risk group might be more complex, involving significant activation or abnormal regulation of B cells, CD8 T cells, and natural killer (NK) cells. Additionally, the hallmark gene set's GSEA analysis (Fig 8F) revealed that the high-risk group's genes were enriched in important pathways such as MYC targets, G2M checkpoint, and E2F targets. Overall, these findings suggest that the identified DEGs may contribute to the regulation of cell cycle, immune response, and carcinogenic processes, emphasizing their potential relevance in disease prognosis and targeted therapeutic approaches (S4 Table).

## PPI network construction

First, we identified the DEGs between the high-risk and low-risk groups. The STRING database was used to build the PPI network of these DEGs. A network diagram was then created by selecting five important model genes and the genes that are directly linked to these five genes using Cytoscape software. We used a Pearson correlation analysis heatmap to display the correlations between these prognostic genes to better investigate their linkages. KIF20A and EZH2 were shown to be strongly correlated (Figs 9A and 9B).

## Gene mutation analysis

After downloading the mutation data from the TCGA dataset, we further investigated the differences in TMB and survival between the high-risk and low-risk subgroups. Unfortunately, despite no significant difference in TMB between the high-risk and low-risk groups, and no clear correlation between TMB and survival (S6 Fig), patients with high TMB in the high-risk group generally exhibited poorer prognosis (Fig 10A). To further analyze the mutation patterns, we generated waterfall plots to show the top 20 genes with the highest mutation frequencies in the TCGA dataset and both risk subgroups (Figs 10B-D). Furthermore, we examined the main mutation types for the risk categories and discovered that the high-risk and low-risk groups were primarily affected by missense mutations, single nucleotide polymorphisms (SNPs), and C>T mutations, respectively (Figs 10E and 10F).

Additionally, we examined the co-occurrence and mutually exclusive associations between the top 20 genes with the highest mutation frequencies in the low-risk group (Fig 10H) and the high-risk group (Fig 10G), as well as the median

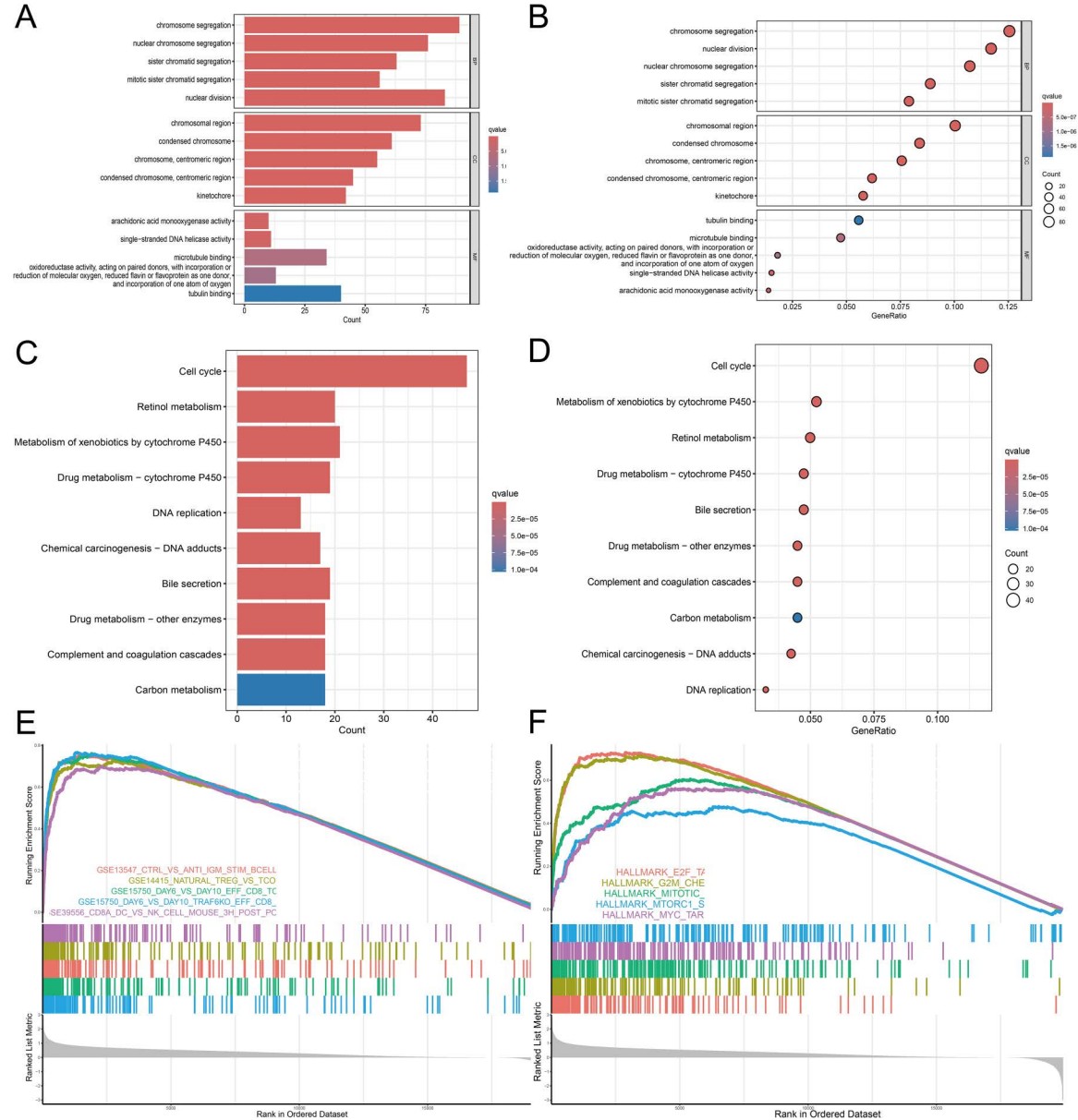

**Fig 8. Functional enrichment analysis of DEGs between risk score subgroups (A, B) shows GO enrichment analysis of DEGs between high-risk and low-risk groups.** (C-D) KEGG enrichment analysis of DEGs between high-risk and low-risk groups. (E) Gene set C7 enrichment analysis (GSEA). (F) Gene set Hallmark enrichment analysis (GSEA).

variation and various mutation types (Figs 10E and F). Different interactions are shown by the colors in the Fig: co-occurrence of gene pairs is shown in green, mutual exclusion is shown in red, and no significant association is shown in white. The level of significance is indicated by the dots or asterisks ($p < 0.05$ or $p < 0.01$).

TP53 and PCLO were found to have a significant co-occurrence association in the high-risk group (Fig 10G) (green, $p < 0.05$). Other gene pairs also showed notable mutation patterns in the low-risk group (Fig 10H). For instance, CTNNB1 and ARID2 were found to have a substantial co-occurrence connection (green, $p < 0.01$).

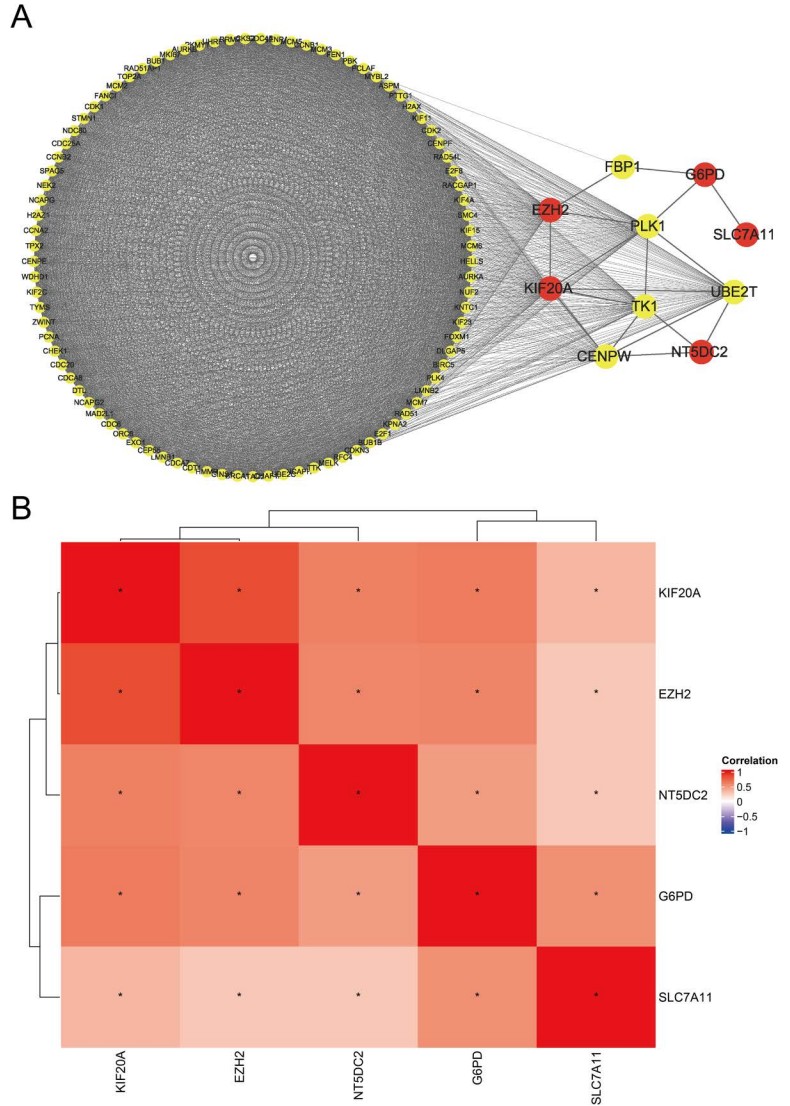

**Fig 9. PPI network and nFRGs Pearson correlation analysis (A) PPI network diagram of nFRGs.** (B) Heatmap of nFRGs correlation.

## Correlation between risk scores and immune microenvironment

The immune cell composition of the tumor microenvironment differed significantly between the high-risk and low-risk groups in the immune infiltration study performed with ssGSEA and CIBERSORT (Figs 11A and 11C). Activated CD4 + T cells, effector memory CD4 + T cells, natural killer T cells, activated dendritic cells, T follicular helper cells, regulatory T cells (Tregs), and M0 macrophages were the most abundant immune response components in the high-risk group. Furthermore, in both analyses, eosinophils for the high-risk group displayed some degree of infiltration. The low-risk group, on the other hand, showed a distinct immune infiltration profile that was mainly enriched in resting mast cells, monocytes, M2 macrophages, activated NK cells, and CD4 + memory T cells.

In terms of immunological function, the high-risk group expressed more MHC class I molecules, whereas the low-risk group displayed increased cytotoxic activity and active interferon responses (Fig 11B).

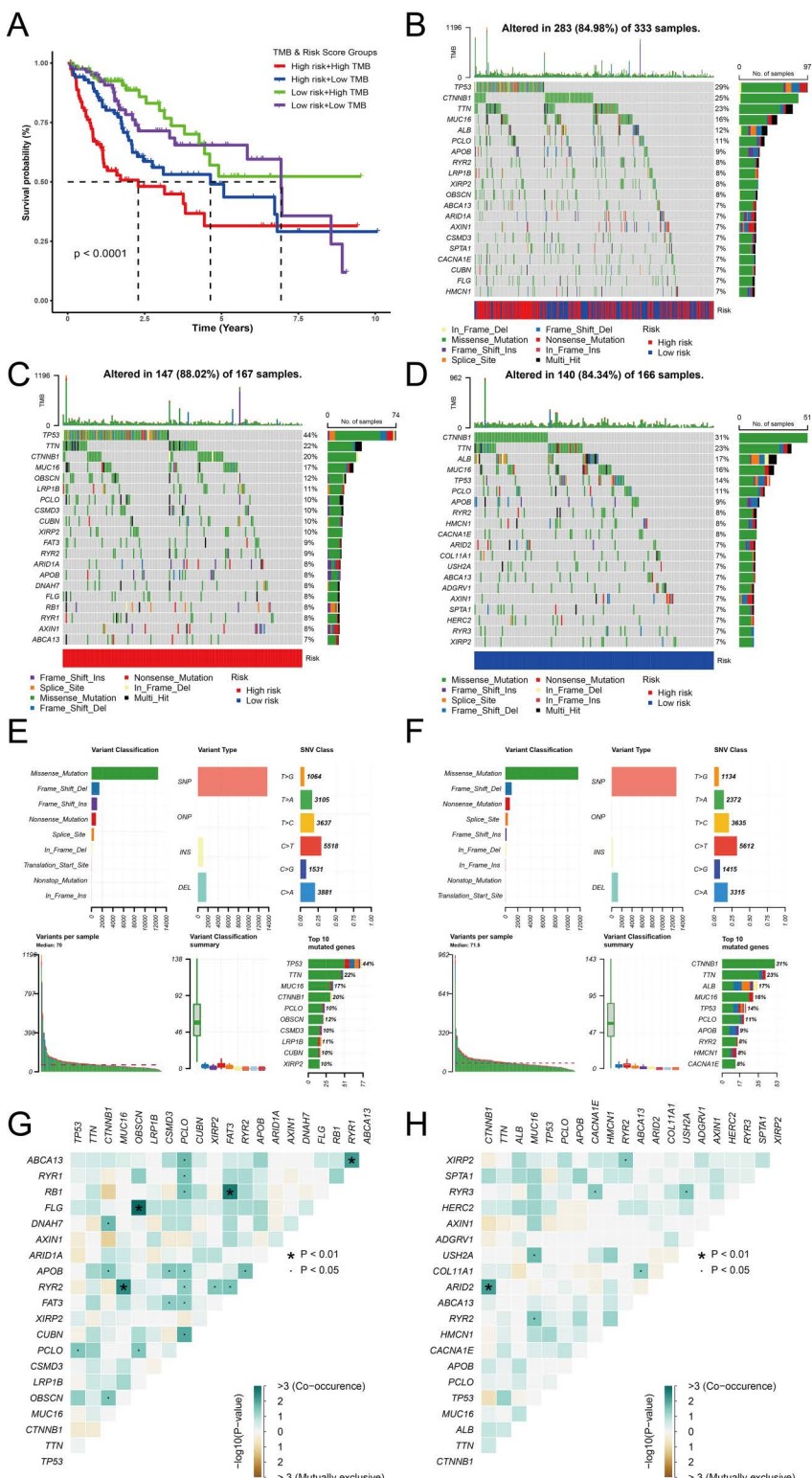

**Fig 10. Tumor mutation analysis (A) The Kaplan-Meier survival curve shows the combined impact of TMB and risk score on survival prognosis.** (B) A waterfall plot showing detailed mutation information for the top 20 genes in the TCGA dataset. (C, D) Waterfall plots showing detailed mutation information for the top 20 genes in the high-risk (C) and low-risk (D) groups. (E, F) Mutation information in the high-risk (E) and low-risk (F) groups categorized by different mutation types. (G, H) Correlation heatmaps of the top 20 mutated genes in the high-risk (G) and low-risk (H) groups.

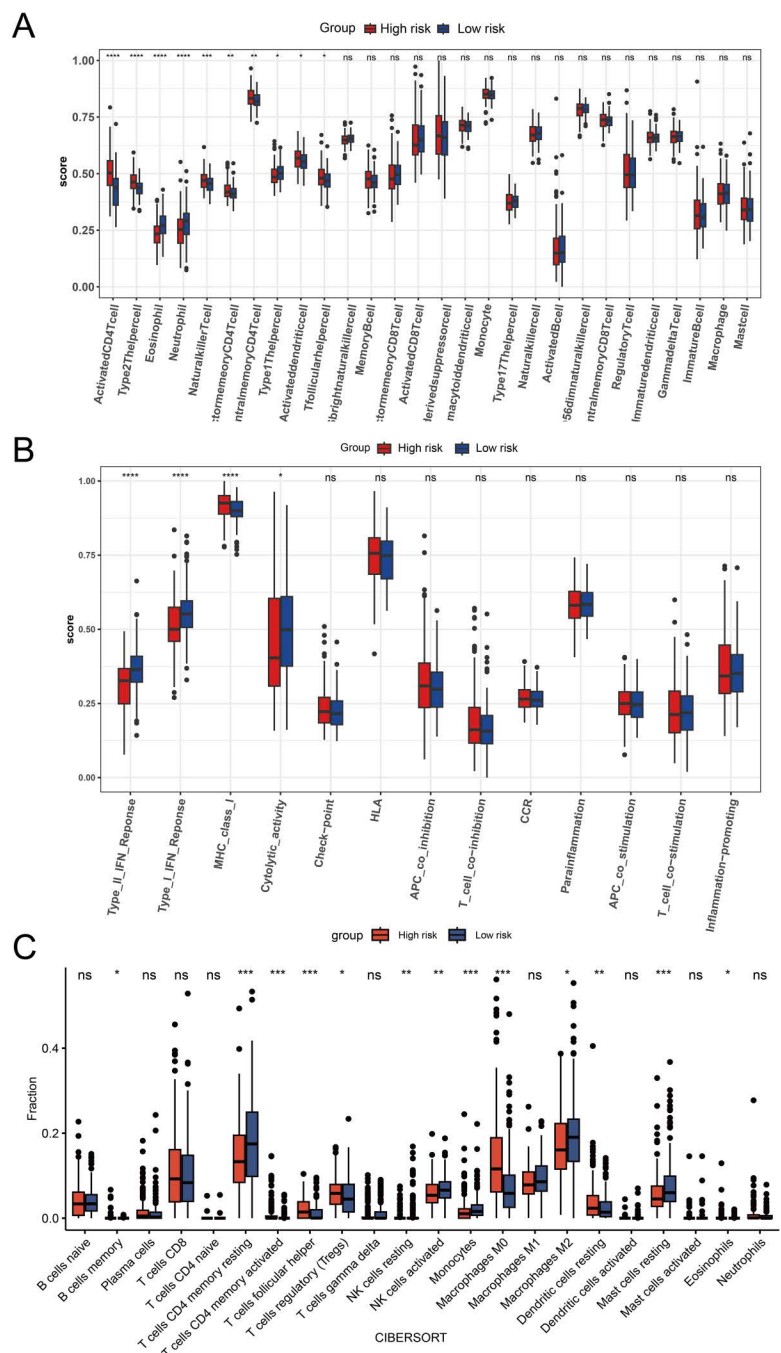

**Fig 11. Immune microenvironment (A) Comparison of immune cell infiltration between the high-risk and low-risk groups using the ssGSEA algorithm.** (B) Comparison of immune function between the high-risk and low-risk groups using the ssGSEA algorithm. (C) Comparison of immune cell infiltration between the high-risk and low-risk groups using the CIBERSORT algorithm.

We examined the relationship between the risk score and infiltration of 22 immune cell types using Spearman's rank correlation coefficient ($\rho$), which is suitable for detecting non-linear associations. Positive correlations were found between the risk score and activated CD4+ memory T cells ($\rho = 0.20$, $p = 1.53e\text{-}04$), follicular helper T cells ($\rho = 0.23$, $p = 1.91e\text{-}05$),

M0 macrophages (ρ=0.37, *p*=8.63e-13), and resting dendritic cells (ρ=0.21, *p*=7.06e-05) (Figs 12B–D). However, no significant correlation was observed with memory B cells (ρ=0.18, *p*=1.05e-03), regulatory T cells (ρ=0.17, *p*=2.11e-03), or eosinophils (ρ=0.16, *p*=1.82e-04) (Figs 12A, 12E and 12F). Resting NK cells showed the strongest negative correlation with the risk score (ρ=−0.20, *p*=1.05e-03) (Figs 12M), while the correlations for other immune cells were weaker (ρ>−0.2) (Figs 12H–L and 12N).

Subsequently, we performed a correlation analysis between the genes in the model and immune infiltrating cells and plotted the corresponding heatmap (Fig 13A). The findings indicate that there is a specific relationship between each gene and immune cells.

### Immune response and immune checkpoints

42 immune checkpoints were found when we screened for common immunological checkpoints that differed significantly across the high-risk and low-risk groups. Only the IDO2 immune checkpoint exhibited an upregulation trend in the low-risk group, while most were upregulated in the high-risk group (Fig 13B).

We discovered a substantial difference in the stromal score between the low-risk and high-risk groups in the TME analysis. TumorPurity, ImmuneScore, and ESTIMATEScore did not change significantly (Fig 13C and S7 Fig). Additionally, the TIDE score analysis showed that the high-risk group's TIDE score was substantially greater than the low-risk group's (Fig 13D).To further explore the impact of risk score grouping on immunotherapy, we analyzed the Immune Prognostic Score (IPS). The results showed that in both the CTLA4-negative and PD1-negative group, as well as the CTLA4-positive and PD1-negative group, the IPS score of the low-risk group was significantly higher than that of the high-risk group.

### Drug sensitivity analysis

We evaluated drug sensitivity by comparing the IC50 values between the high-risk and low-risk groups to find possible medications for the HCC prognostic model (Fig 14). According to the findings, there were notable variations among 103 medications. According to the data, people in the high-risk group were more sensitive to the majority of medications.

### Drug prediction based on nFRGs

We compiled the significant correlations between each medicine and the associated genes after choosing the top ten candidate drugs based on the examination of the relationship between distinctive genes and drugs (Table 3 and S5 Table). Among these, acetaminophen (CTD 00005295) and 7646-79-9 (CTD 00000928) have P-values of 3.77E-04 and 8.84E-05, respectively, and are strongly linked to five distinctive genes (KIF20A, NT5DC2, G6PD, SLC7A11, and EZH2).

## Discussion

In this study, we used univariate Cox regression, LASSO regression, and multivariate forward stepwise Cox regression to identify five genes—G6PD, NT5DC2, SLC7A11, KIF20A, and EZH2—as independent predictive characteristics for HCC. These characteristics successfully differentiated between high-risk and low-risk patients and showed strong predictive power in both the training and testing groups, indicating that high-risk patients have a lower OS. Additionally, The GEO database (GSE14520 dataset) was used as an external validation set to verify the reliability of the constructed predictive biomarkers.validation results from the GEO database supported these genetic characteristics' potential as predictive biomarkers for HCC by confirming their accuracy and dependability.

Furthermore, this study is the first to demonstrate the originality and predictive power of the model we built (G6PD, KIF20A, EZH2, NT5DC2, SLC7A11), as well as to compare and assess several published FRGs prognostic models. According to the results, our model outperformed the majority of previously published models in every performance indicator, including average C-index, validation set C-index, and AUC values at 1, 2, and 3 years. In the testing set, several

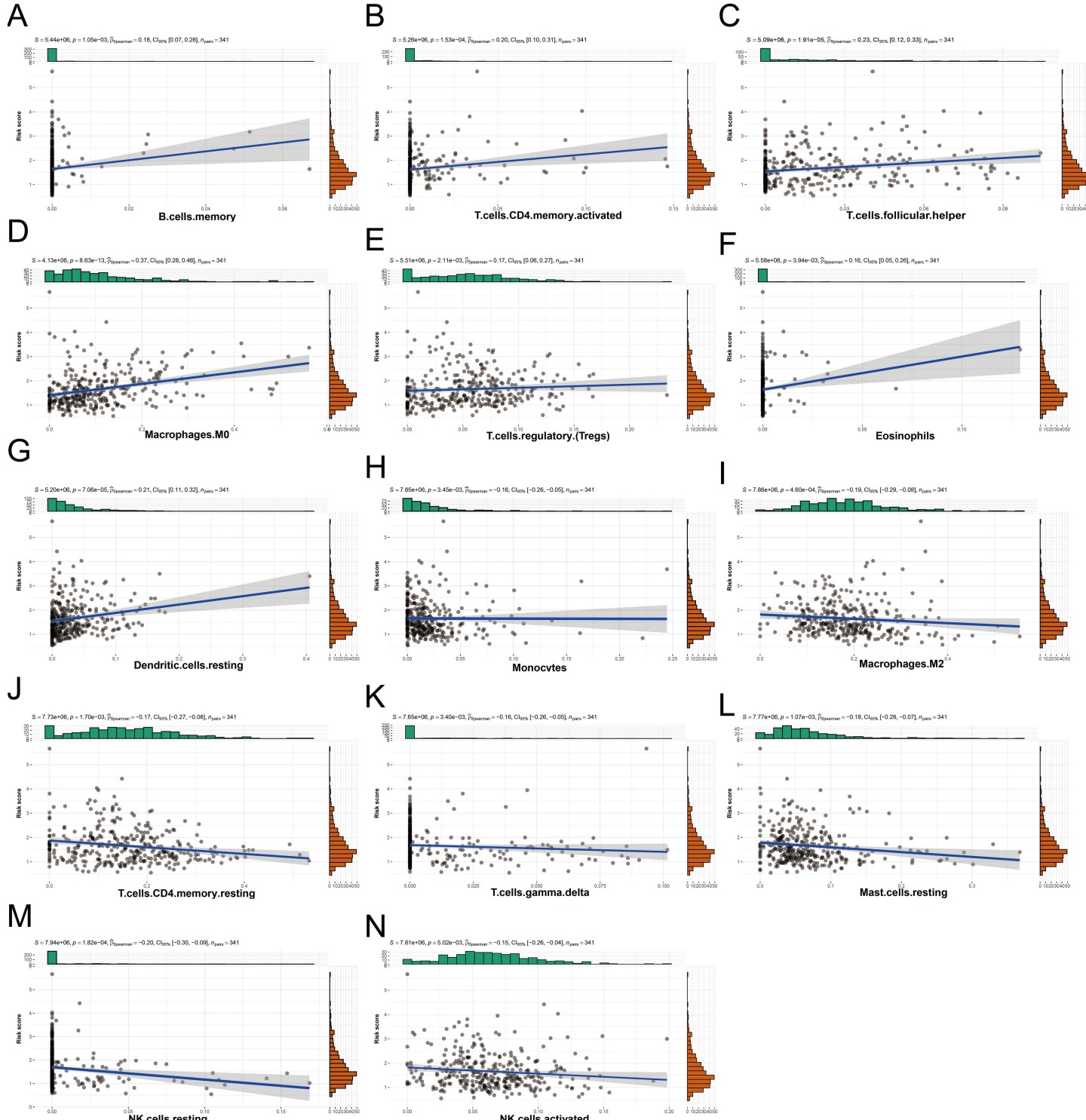

**Fig 12. Correlation between risk score and immune cells (A-N) The relationship between risk score and immune cell infiltration abundance.**

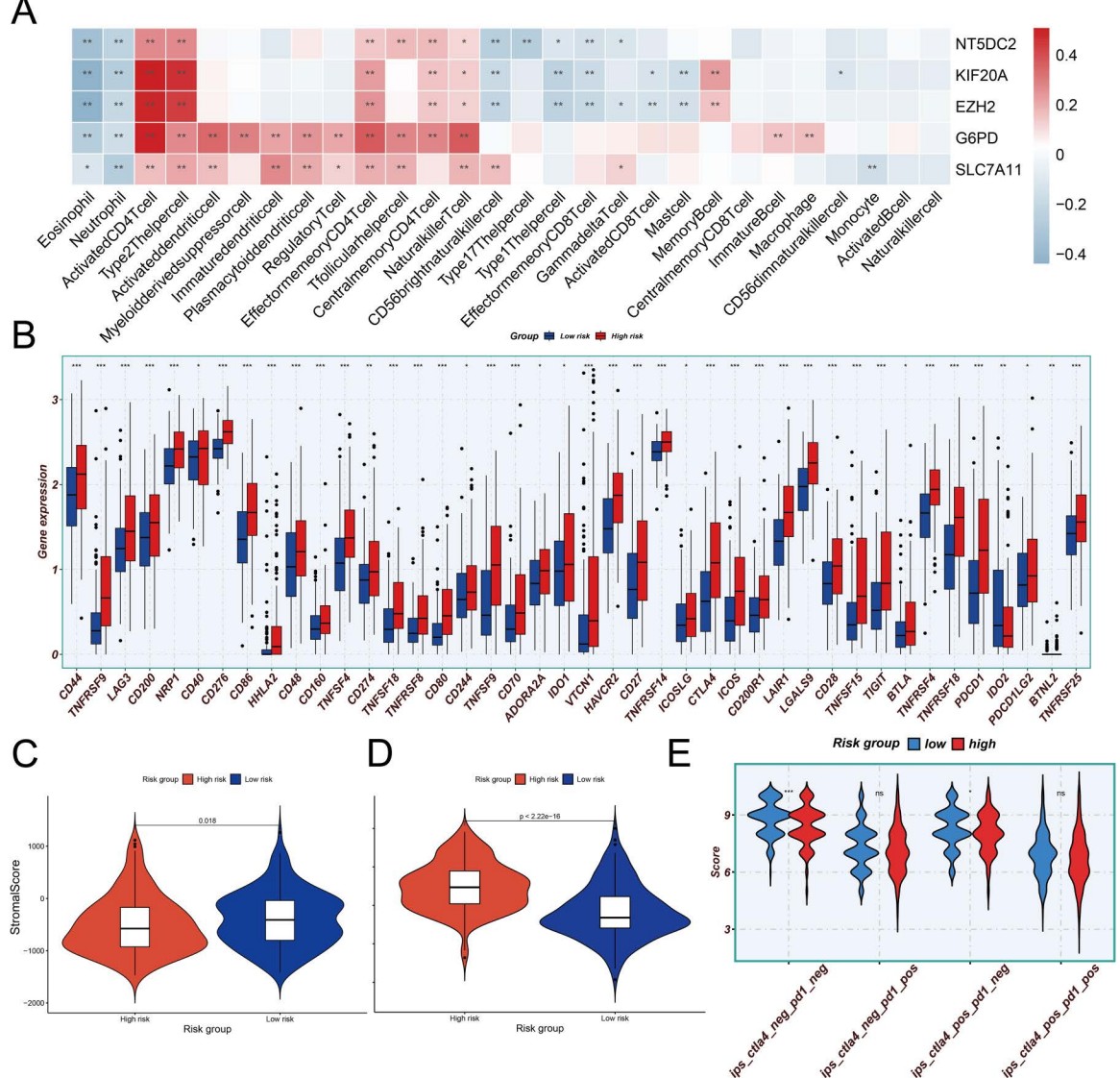

**Fig 13. The relationship between nFGRs and their risk subgroups in the immune environment and immune therapy response.** (A) Correlation analysis of nFGRs and immune infiltrating cells; (B) Comparison of immune checkpoint expression between the high-risk and low-risk groups; (C) Differences in StromalScore between the high-risk and low-risk groups; (D) Differences in TIDE scores between the high-risk and low-risk groups. (E) Differences in IPS between the high-risk and low-risk groups.

models had comparatively poor generalization ability, even though all models performed well in the training set. Our model's consistent performance over several datasets indicates its potential for real-world use, indicating that future model development should concentrate on generalization skills to guarantee the model's dependability on separate datasets.

However, the assumptions and limitations of the statistical analyses and models warrant further discussion. First, both the univariate and multivariate Cox regression models are based on the proportional hazards assumption, which assumes that the hazard ratios between different groups remain constant over time. However, this assumption may not hold in certain situations. Although we performed thorough data preprocessing and checks, there remains a risk that this assumption may be violated in some cases, which could impact the interpretation of the results. Furthermore, while LASSO regression

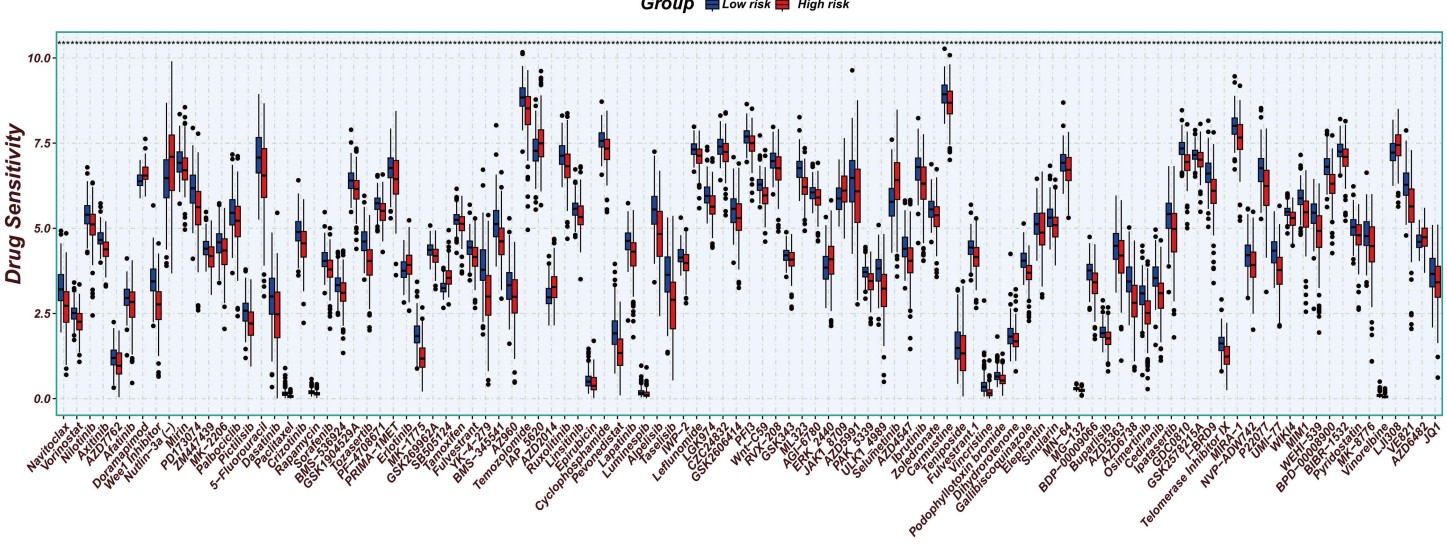

**Fig 14. Comparison of drug sensitivity differences between the high-risk and low-risk groups.**

**Table 3. Candidate drug predicted using DSigDB.**

| Drug names | P-value | Adjusted P-value | Genes |
|---|---|---|---|
| gossypol HL60 DOWN | 5.61E-05 | 0.019703855 | NT5DC2;KIF20A |
| 7646-79-9 CTD 00000928 | 8.84E-05 | 0.019703855 | G6PD;NT5DC2;KIF20A;SLC7A11;EZH2 |
| COPPER CTD 00005706 | 2.36E-04 | 0.033631487 | G6PD;NT5DC2;KIF20A;SLC7A11 |
| Glycidamide CTD 00002776 | 3.18E-04 | 0.033631487 | G6PD;SLC7A11 |
| acetaminophen CTD 00005295 | 3.77E-04 | 0.033631487 | G6PD;NT5DC2;KIF20A;SLC7A11;EZH2 |
| doxorubicin CTD 00005874 | 4.96E-04 | 0.03607901 | G6PD;KIF20A;EZH2 |
| cinnamaldehyde CTD 00000671 | 6.66E-04 | 0.03607901 | G6PD;SLC7A11 |
| bicalutamide CTD 00002279 | 7.06E-04 | 0.03607901 | KIF20A;SLC7A11 |
| Enterolactone CTD 00001393 | 0.001059739 | 0.03607901 | KIF20A;SLC7A11;EZH2 |
| LUCANTHONE CTD 00006227 | 0.001105295 | 0.03607901 | KIF20A;EZH2 |

is effective in reducing the number of variables and preventing overfitting, it may also lead to the exclusion of some potentially significant genes. Therefore, future studies should consider employing additional model validation techniques, such as cross-validation or external validation datasets, to further confirm the robustness and stability of the model.

Despite the excellent performance of all models in both the training and testing sets in this study, we also observed that certain previously published models exhibited poor generalization ability in the testing set, particularly in the context of clinical applications. This suggests that current predictive models may be prone to overfitting, highlighting the need to focus on enhancing the generalizability of the models during their development and optimization, ensuring their effectiveness in different datasets and clinical settings. Moreover, while our model outperformed existing FRG prognostic models across several indicators, the selection and weighting of these indicators may be influenced by the characteristics of the dataset and the analysis methods, potentially limiting the generalizability of the findings. Therefore, future studies should validate our findings in a broader range of external datasets to determine whether our results are applicable to a wider HCC patient population.

Another limitation is that we chose to exclude samples with a total survival time (OS) of less than 30 days to minimize the potential confounding effects of early non-tumor-related mortality on the analysis. Additionally, to ensure the reliability of the statistical analysis, we excluded samples with incomplete clinical data. We acknowledge that this complete-case analysis may introduce selection bias, but given the nature and proportion of missing data, we believe that directly applying multiple imputation methods could introduce additional uncertainty, which may impact the robustness of the results. Therefore, to address potential selection bias, we have minimized the impact of missing data as much as possible during the study design phase and conducted strict quality control in the statistical analysis.

Additionally, Another limitation is that our study primarily relied on data from public databases, which may suffer from sample selection bias or lack certain clinical information. Future research incorporating data from multi-center clinical samples would help to enhance the clinical applicability of the model. Additionally, although our study indicates that the selected genes are closely associated with the prognosis of HCC, how these genes interact in the tumor microenvironment and their specific mechanisms remain unclear. Therefore, future investigations should focus on further exploring the molecular mechanisms of these genes, particularly their roles in the liver cancer immune microenvironment.

In summary, while this study provides a robust statistical model for prognostic evaluation in HCC and validates the potential biomarkers associated with HCC prognosis, the assumptions of the statistical methods and the limitations of the models still need to be addressed and improved in future research. Specifically, validating the stability and generalizability of the models and addressing the challenges they may face in clinical applications will be crucial directions for future studies.

Ferroptosis-related pathways are crucial to the development of HCC, according to earlier research [25–27]. By controlling gene expression, cancer cells can develop resistance to ferroptosis [28], and the five FRGs that we found are important for both tumor growth and ferroptosis. G6PD (glucose-6-phosphate dehydrogenase) is a key enzyme in the pentose phosphate pathway and has been shown to inhibit ferroptosis in HCC by regulating the reductase (POR). POR is upregulated in G6PD deficiency, which prevents HCC cells from growing and spreading [29]. Additionally, G6PD expression is strongly linked to OS and the immune microenvironment in HCC, and its overexpression lowers ferroptosis in HCC, underscoring its potential as a tumor prognostic and therapeutic biomarker [30]. Increased expression of NT5DC2 (5'-nucleotidase domain-containing 2) has been linked to immunological infiltration and is a predictor of a poor outcome in patients with HCC, with possible uses in clinical stratification [31–34]. By controlling intracellular iron homeostasis and oxidative stress reactions, NT5DC2 contributes significantly to ferroptosis. NT5DC2 plays a crucial function in the regulation of ferroptosis, as evidenced by the fact that its knockdown increases cell susceptibility to ferroptosis inducers [35]. As a component of the system Xc^-, SLC7A11 (solute carrier family 7 member 11) facilitates glutathione synthesis and is in charge of cysteine transport, both of which are critical for antioxidant defense [26]. By controlling glutathione synthesis, SLC7A11 prevents ferroptosis and lowers lipid peroxide buildup. Research has demonstrated that SLC7A11 inhibits ferroptosis, which increases tumor growth in addition to HCC cell proliferation [36–40]. By controlling microtubule dynamics and cell division, the protein KIF20A (Kinesin Family Member 20A) plays a crucial role in HCC proliferation and tumor growth by facilitating chromosomal movement during cell division [41]. The transcription factor Gli2, which is a downstream target gene of the Hedgehog (Hh) signaling pathway, activates the FoxM1-MMB complex to enhance KIF20A expression, which in turn stimulates the formation of tumors and the proliferation of HCC cells. High KIF20A expression is strongly linked to HCC recurrence and a poor prognosis for patients, according to clinical studies [42]. Additionally, KIF20A interacts with the c-Myc pathway and immune evasion mechanisms to contribute to the development of HCC and the immunological response to treatment. By enhancing the benefits of anti-PD-1 immunotherapy, targeting KIF20A therapy may open up new avenues for precision medicine and immunotherapy in HCC [43]. Histone methyltransferase EZH2 (Enhancer of Zeste Homolog 2) mostly suppresses gene expression by trimethylating H3K27. In addition to controlling the expression of TFR2 (Transferrin Receptor 2) through epigenetic pathways, EZH2 is essential for the development, growth, metastasis, and immune evasion of HCC. This prevents ferroptosis and increases drug resistance in HCC cells [44]. As a

result, EZH2 targeting could be a crucial approach to treating HCC [45]. The existing research has not disclosed the direct mechanistic relationship between EZH2 and KIF20A, even though we discovered a strong association between the two in the PPI network diagram and Pearson correlation analysis. However, research has revealed that EZH2 and KIF20A are strongly linked to the growth and prognosis of cancer and coexist in many prognostic indicators [46–50], offering crucial hints for further investigation.

After describing the individual roles of the ferroptosis-related genes (G6PD, KIF20A, EZH2, NT5DC2, SLC7A11), it is essential to investigate how these genes might interact with each other to influence the tumor microenvironment (TME) in a synergistic manner. These genes likely regulate ferroptosis through multiple interconnected mechanisms, where each gene not only plays a role in its specific pathway but also interacts with others to fine-tune the overall process. Their combined actions can influence critical biological processes such as iron homeostasis, oxidative stress, and antioxidant defense, which are all key factors in ferroptosis induction.

For instance, G6PD, as an important regulator of the pentose phosphate pathway, may impact the cellular oxidative stress balance, potentially enhancing or suppressing ferroptosis depending on the metabolic context. Meanwhile, SLC7A11, which is involved in glutathione synthesis and antioxidant defense, can work in conjunction with G6PD to modulate cellular resistance to ferroptosis by maintaining the cellular antioxidant capacity. KIF20A, which is linked to cell division, might influence the proliferation of tumor cells, thereby modulating their susceptibility to ferroptosis. The upregulation of KIF20A could potentially enhance the evasion of ferroptosis, promoting tumor growth and progression.

EZH2, a key epigenetic regulator, could influence the expression of other ferroptosis-related genes by altering chromatin structure, which might further contribute to ferroptosis resistance or enhance susceptibility. Similarly, NT5DC2's role in regulating immune responses and its potential to influence ferroptosis suggests it might play a part in modulating the immune microenvironment. By interacting with immune cells, these genes could either support immune evasion or enhance immune responses, depending on their expression levels and activity.

The interaction of these genes creates a complex relationship between tumor progression, ferroptosis, and immune modulation. Some genes may suppress ferroptosis to evade immune detection, while others may induce ferroptosis to enhance immune recognition and tumor cell clearance. Understanding how these ferroptosis-related genes modulate the immune microenvironment and contribute to immune escape mechanisms is crucial for the development of new immunotherapies that target these pathways to promote ferroptosis in tumor cells and enhance anti-tumor immunity.

Our enrichment analysis revealed that the differentially expressed genes (DEGs) between the high- and low-risk groups are significantly enriched in key biological processes such as cell division, chromosome segregation, cell cycle, and mitosis, which are closely related to tumor proliferation and metastasis [51–54]. These results suggest that genes like G6PD may play a role in ferroptosis by regulating cell proliferation and division processes, further influencing tumor growth and metastasis.

Specifically, in terms of cellular components, the DEGs are enriched in centromeric and chromosomal regions, indicating that they may be involved in ferroptosis through the regulation of the cytoskeleton. For example, the role of KIF20A is closely related to the regulation of microtubule dynamics, and its key function in cell division may influence ferroptosis by regulating iron homeostasis and oxidative stress responses [41]. Additionally, SLC7A11 controls antioxidant defense by promoting the transport of cysteine and the synthesis of glutathione, thereby inhibiting ferroptosis. These processes are closely associated with the cell cycle and proliferation [26]. By regulating these cellular biological processes, these genes may exacerbate tumor proliferation and drug resistance in the high-risk group.

KEGG analysis shows that the DEGs in the high-risk group are enriched in key processes such as the cell cycle, carbon metabolism, and DNA replication, suggesting that these genes are closely related to cell proliferation and metabolic imbalance [55–57]. Notably, the role of EZH2 involves the epigenetic regulation of TFR2 expression, which further inhibits ferroptosis and enhances drug resistance [44], a process closely related to cell cycle regulation and tumor proliferation

and metastasis. Therefore, the enrichment of these genes in the high-risk group may influence the occurrence of ferroptosis and promote tumor progression by altering iron homeostasis, oxidative stress, and cell cycle regulation.

GSEA analysis further revealed that the immune microenvironment of patients in the high-risk group is complex and closely related to immune evasion mechanisms [58–60]. This provides new insights into the role of these genes in the regulation of ferroptosis. For example, the expression of NT5DC2 is closely associated with immune infiltration and plays a crucial role in immune evasion [31–34], which may affect the response of patients to immunotherapy. Moreover, the interaction of KIF20A with the c-Myc pathway and immune evasion mechanisms may also be associated with both immune evasion and ferroptosis [42]. Additionally, our Hallmark gene set analysis revealed that genes in the high-risk group are enriched in pathways such as MYC target genes, G2M checkpoint, and E2F target genes, which are crucial for tumor progression, cell cycle regulation, and proliferation [61–63].

In summary, our enrichment analysis results indicate that DEGs in the high- and low-risk groups are closely related to the cytoskeleton, lipid metabolism, chromosomal regulation, proliferative processes, and the cell cycle. These genes are also associated with immune evasion strategies and a complex immune microenvironment. These findings provide further insights into the interactions of these ferroptosis-related genes and their potential in regulating ferroptosis within the tumor microenvironment (TME), as well as their viability as therapeutic targets for ferroptosis. Future studies can further explore the interactions between these genes, particularly in the TME, to understand how they synergistically regulate the occurrence of ferroptosis and provide a theoretical basis for the development of individualized treatment strategies.

It is imperative to investigate further how these findings might be used in therapeutic techniques, given the correlation found in our work between the DEGs in the high-risk group and the immunological microenvironment and immune evasion mechanisms. Immunotherapy has emerged as a novel therapeutic approach that has improved survival rates for HCC patients globally [64]. Immune checkpoint clinical research is comparatively advanced and well-established. Over time, immune checkpoint modification has gained popularity as an immunotherapy technique [65]. Furthermore, according to recent research, tumor mutational burden, or TMB, may be a predictive biomarker for immunotherapy, and a high TMB may indicate a favorable reaction to immune checkpoint inhibitors (ICIs) [66]. Initially, we examined the gene mutations in the prognostic genes in HCC and the gene mutations in the low-risk and high-risk groups. TMB did not significantly affect the survival rate, and there was no statistically significant difference in TMB between the high-risk and low-risk groups, according to the results. However, we discovered that missense mutations were prevalent, with the two main altered genes in both groups being TP53 and CTNNB1. It is commonly recognized that many types of cancer include mutations in TP53 and CTNNB1. It has been demonstrated that the tumor suppressor gene TP53 controls the metabolism of tumor cells [67,68].TP53-mutant HCC tissues, on the other hand, are often poorly differentiated and can be linked to angiogenesis and vascular invasion [69]. The protein β-catenin, encoded by the CTNNB1 gene, has been shown to encourage immune evasion and may influence how well HCC patients respond to immunotherapy [70]. Additionally, a deeper comprehension of the immune milieu will facilitate the creation of novel HCC treatment approaches.

The two patient groups based on risk scores differ significantly in the number of immune cells, according to the immune infiltration study. We discovered that the proportion of M0 macrophages, Tregs, and natural killer T (NKT) cells was higher in the high-risk group. Inhibiting anti-tumor immunity and influencing NK cell activity, Tregs and NKT cells are important components of the tumor microenvironment of HCC, contributing to tumor growth [71]. M0 macrophages, which are strongly linked to a poor prognosis, increased angiogenesis, and a low response to immunotherapy, also contribute significantly to the development of tumors in HCC. This discovery offers fresh approaches to risk assessment and tiered treatment in HCC [72]. The aforementioned findings imply that an immunosuppressive environment may be the cause of the high-risk group's poor prognosis. Furthermore, immunotherapy based on ICIs has a lot of potential for treating HCC. We compared the two risk groups' immunological checkpoints. ICIs may be more beneficial for low-risk patients, as the high-risk group exhibited greater expression of 41 immune checkpoints, indicating a stronger immune suppression status. Additionally, we discovered that low-risk patients had lower TIDE

ratings, suggesting that immunotherapy may work better for them. Furthermore, we further evaluated immunotherapy using the IPS and found that the IPS of the low-risk group were generally higher than those of the high-risk group. In addition, the IPS of the low-risk group were significantly higher than those of the high-risk group in both the CTLA4-negative and PD1-negative group and the CTLA4-positive and PD1-negative group, further confirming the advantage of the low-risk group in immunotherapy.By analyzing the TIDE and IPS scores, clinicians can predict a patient's response to immunotherapy based on their immune profile, thus providing a basis for personalized treatment. We found that the low-risk group exhibited more favorable immune responses in both TIDE and IPS scores, suggesting that these patients may respond better to immunotherapy, especially ICIs. This model helps clinicians identify patients who are likely to benefit from immunotherapy, avoiding ineffective treatment for low-responding patients, thereby improving treatment efficacy and reducing side effects.

Moreover, based on risk scores, clinicians can adjust treatment strategies. High-risk patients may require combination therapy, such as chemotherapy or targeted therapy, while low-risk patients may benefit from ICIs alone. Given the better response of the low-risk group to immunotherapy, ICIs may be the preferred treatment for this population. This model can also provide precise prognostic assessment and personalized follow-up management for patients, ensuring that high-risk patients receive more frequent monitoring and proactive intervention.

Finally, the immune scoring model can provide guidance for clinical trial design, helping to refine inclusion criteria, ensuring the efficacy and safety of immunotherapy, and serving as an efficacy biomarker to evaluate new drugs or treatment protocols. By integrating this model into clinical practice, it allows for tailored treatment plans, optimizing immunotherapy outcomes and improving prognosis, which holds significant clinical value, especially in the context of using ICIs in immunotherapy.

However, there are several limitations to the clinical application of this immune scoring model. One significant limitation is the lack of detailed treatment data in the public databases such as TCGA and GSE14520. These databases do not provide information about the specific treatments patients received, such as surgery, chemotherapy, or immunotherapy regimens. As a result, the model may conflate true prognostic effects (inherent tumor biology) with treatment-related factors (such as access to specific therapies), potentially leading to biased conclusions.

For instance, high-risk patients may have been ineligible for curative therapies or have received less aggressive treatments, which could affect their prognosis independently of the tumor's biological behavior. This lack of treatment data introduces unmeasured confounding, limiting the model's ability to fully assess the impact of treatment on patient outcomes. Future studies should aim to integrate treatment data, including detailed information about specific therapies and their effects on the tumor microenvironment, to refine this model and enhance its clinical applicability. Incorporating treatment-related variables will help differentiate the prognostic value of the immune scoring model from treatment-related biases, providing a clearer picture of how immune profiles influence treatment response.

We also used the prognostic model to assess the sensitivity to targeted medications and chemotherapy to forecast the treatment results of the patients. The findings indicated that the majority of medications were more sensitive in the high-risk group, indicating that these treatments would work better for high-risk individuals.

We also used DsigDB to screen the association between feature genes and medications to improve the therapy of nFRGs, and this revealed some promising therapeutic possibilities. By looking at the relationship between these medications and important genes, we discovered that several medications, including acetaminophen (CTD 00005295) and 7646-79-9 (CTD 00000928), exhibited strong connections with the feature genes. The five feature genes (KIF20A, NT5DC2, G6PD, SLC7A11, and EZH2) were substantially correlated with these two medications. The aforementioned medications may provide viable therapeutic approaches by modifying the expression of these genes, given their possible significance in tumor growth and therapy response. Additional research could examine how these medications work in particular pathological situations and assess their viability for clinical use

## Conclusion

In conclusion, this research aims to determine five important genes with prognostic predictive value and investigate the possible contribution of ferroptosis-related gene features (nFRGs) to the prognosis of hepatocellular carcinoma (HCC). Our findings imply that low-risk patients might benefit more from immunotherapy, including immune checkpoint inhibitors (ICIs), as they exhibit lower immune suppression and better immune responsiveness. Conversely, the high-risk group shows increased expression of immune checkpoints, indicating a stronger immune-suppressive status.However, high-risk patients have greater sensitivity to targeted medications and chemotherapy, suggesting that they may respond better to these treatments. However, this study has certain limitations. As our analysis primarily relies on bioinformatics analysis using public datasets, the lack of in vitro or in vivo experimental validation of the selected genes limits the clinical applicability of the results. Nevertheless, based on the results from multidimensional data analysis and cross-validation, we have provided preliminary support for the potential application of the nFRGs model in HCC. Future experimental studies will help confirm the specific roles of these five selected genes in ferroptosis and hepatocellular carcinoma, and further validate their clinical translational value.

## Supporting information

**S1 Fig. Immunohistochemical expression of novel ferroptosis-related genes in normal and HCC tissues from the HPA database.**
(TIF)

**S2 Fig. The AUC values of each model at 1 year, 2 years, and 3 years.**
(TIF)

**S3 Fig. Kaplan-Meier survival analyses based on diagnosis year and risk stratification.**
(TIF)

**S4 Fig. Forest plots: Univariate Cox analysis for risk score subgroups and clinical features in the TCGA and GSE14520 datasets.**
(TIF)

**S5 Fig. The AUC values of our prognostic model compared with other ferroptosis-related prognostic models at 1 year, 2 years, and 3 years in each dataset.**
(TIF)

**S6 Fig. Tumor Mutation Burden (TMB) analysis.**
(TIF)

**S7 Fig. Comparison of risk subgroups in the tumor microenvironment (TME).**
(TIF)

**S1 Table. Intersection of ferroptosis and upregulated differential genes.**
(XLSX)

**S2 Table. Pairwise comparison of overall survival among combined diagnosis year and risk group categories using Log-rank test.**
(XLSX)

**S3 Table. Previously published ferroptosis-related prognostic models.**
(XLSX)

**S4 Table. Functional analysis of differentially expressed genes between high- and low-risk subgroups.**
(XLSX)

**S5 Table. Drug prediction for novel ferroptosis-related genes (nFRGs).**
(XLSX)

## Acknowledgments

We sincerely thank the Huang Jingjing research group members for providing resources and support during the study. We also extend our gratitude to The Cancer Genome Atlas (TCGA) and the GEO database for their publicly available data, which were crucial for our analysis—special thanks to Professor Huang Jingjing for her valuable guidance in statistical analysis and bioinformatics. Additionally, we appreciate the technical assistance provided by the Guangxi Key Laboratory of Translational Medicine for Treating High-Incidence Infectious Diseases with Integrative Medicine and the constructive comments from the reviewers, which helped improve the quality of this manuscript.

## Author contributions

**Data curation:** Yu Zhang, Yuanqin Du.

**Investigation:** Chengting Wu, Xinyuan Chen.

**Methodology:** Chengting Wu, Xinyuan Chen.

**Software:** Jian Xu.

**Validation:** Yujiao Peng, Lu Lu.

**Visualization:** Lu Lu.

**Writing – original draft:** Chengting Wu, Xinyuan Chen.

**Writing – review & editing:** Jingjing Huang, Hongna Huang.

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
