## [Decision Letter · Decision Letter 0]

16 Jan 2025

Dear Dr. Huang,

Thank you for submitting your manuscript to PLOS ONE. After careful consideration, we feel that it has merit but does not fully meet PLOS ONE’s publication criteria as it currently stands. Therefore, we invite you to submit a revised version of the manuscript that addresses the points raised during the review process.

Please respond to reviewers' comments individually.

We look forward to receiving your revised manuscript.

Kind regards,

Xiaosheng Tan

Academic Editor

PLOS ONE

Journal Requirements:

“This research was funded by the National Natural Science Foundation of China grant number [No. 82460957], the Guangxi Natural Science Foundation Project [No. 2022GXNSFAA035460, No. 2024GXNSFDA010005], and the Guangxi Graduate Education Innovation Program [No. YCSW2023395].”

5. Please remove all personal information, ensure that the data shared are in accordance with participant consent, and re-upload a fully anonymized data set. Note: spreadsheet columns with personal information must be removed and not hidden as all hidden columns will appear in the published file. Additional guidance on preparing raw data for publication can be found in our Data Policy (https://journals.plos.org/plosone/s/data-availability#loc-human-research-participant-data-and-other-sensitive-data) and in the following article: http://www.bmj.com/content/340/bmj.c181.long .

Reviewers' comments:

Reviewer's Responses to Questions

**Comments to the Author**

1. Is the manuscript technically sound, and do the data support the conclusions?

Reviewer #1: Yes

Reviewer #2: Yes

Reviewer #3: No

2. Has the statistical analysis been performed appropriately and rigorously?

Reviewer #1: Yes

Reviewer #2: Yes

Reviewer #3: I Don't Know

3. Have the authors made all data underlying the findings in their manuscript fully available?

Reviewer #1: Yes

Reviewer #2: Yes

Reviewer #3: Yes

4. Is the manuscript presented in an intelligible fashion and written in standard English?

Reviewer #1: Yes

Reviewer #2: Yes

Reviewer #3: Yes

Reviewer #1: The authors aim to develop a novel ferroptosis-related gene signature and provide extensive empirical evidence for the development of the FRG-based risk model. They assess its predictive power to differentiate high-risk and low-risk patients, demonstrating strong predictive accuracy in both training and testing groups. Additionally, the authors perform GO, KEGG, and GSEA analyses to elucidate the differences in biological functions and molecular mechanisms between the differentially expressed genes (DEGs) in high-risk and low-risk groups. Furthermore, gene mutation, drug sensitivity analysis, and drug predictions offer valuable insights for future clinical therapy.

Overall, this is a well-written manuscript that requires only minor revisions.

1. First, the formatting of the p-values is inconsistent; please ensure they are standardized throughout.

Line 214 limma, survival, and caret. The criteria for statistical significance were P < 0.05,

Line 329 following symbols: ** for p < 0.01, and * for p < 0.05.

Line 332 at p < 0.01, especially in the TCGA

Line 439 the high-risk group (Figure 10G) (green, P < 0.05). Other gene pairs also showed

Line 441 and ARID2 were found to have a substantial co-occurrence connection (green, P <

2. The text contains inconsistencies in formatting and grammatical errors that need to be addressed.

Line 266 Figure 3. Model construction and validation (A)Based on multiple machine learning algorithms.

Line 466 Figure 11. Immune Microenvironment(A) Comparison of

Line 467 high-risk and low-risk groups using the ssGSEA algorithm.(B) Comparison of immune function

Line 468 between the high-risk and low-risk groups using the ssGSEA algorithm.(C) Comparison of

Line 662 HCC, contributing to tumor growth [72].M0 macrophages, which are strongly linkeds

Line Figure S2 The AUC values of each model at 1 year, 2 years, and 3 years.AUC Area Under Curve��HR Hazard Ratio

3. Streamline the description of Figure 12 for clarity and conciseness.

4. Figures 8, 9, 10, and 12 become difficult to discern when enlarged and may require reformatting or replacement with higher-resolution images.

5. Supplement Figure 1 by including immunohistochemistry images with a scale bar and clear indicator markings.

Reviewer #2: This manuscript presents a comprehensive study developing a novel ferroptosis-related gene signature (nFRGs) for hepatocellular carcinoma (HCC). The authors identified and validated a five-gene signature (KIF20A, NT5DC2, G6PD, SLC7A11, and EZH2) using data from TCGA and GSE14520 datasets. They demonstrated that this signature effectively stratifies patients into high-risk and low-risk groups with distinct prognostic outcomes, immune characteristics, and treatment sensitivities. The model showed superior predictive performance compared to existing ferroptosis-related signatures and provided insights into personalized treatment strategies for HCC patients.

1. Authors have used the HCC patients' RNA sequencing and related clinical data from the TCGA database. However, the RNA sample's origin, patient demographics (age, sex, etc.), and detailed clinical characteristics are missing from the manuscript. These should be added to provide essential context.

2. Using the median as a cutoff is overly simplistic and may not accurately reflect biological or clinical differences. Consider add new grades like “low, intermediate, high risk”

3. Several instances of incorrect parentheses format appear in the manuscript, particularly in the abstract section. For example: "Tumor Immune Dysfunction and Exclusion�TIDE�scores" uses Chinese parentheses instead of standard ones. These formatting inconsistencies should be standardized throughout.

4. The manuscript would benefit from expanded discussion of the potential mechanisms linking these five identified genes to ferroptosis regulation. More detailed exploration of the interaction between these genes and their roles in the ferroptotic pathway is needed.

5. The manuscript should include more detailed discussion of statistical assumptions and limitations to help readers better understand the strengths and potential constraints of the findings. Important considerations such as validation requirements and generalizability should be addressed.

Reviewer #3: The manuscript presents a novel ferroptosis-related gene signature (nFRGs) for predicting prognosis, immune characteristics, and treatment responses in hepatocellular carcinoma (HCC). However, it suffers from significant lack of clarity, and low-quality data presentation. The concerns outlined below highlight major and minor issues that must be addressed to improve the manuscript.

Major Concerns

Low Figure Quality

Except for Figures 1, 2, and 7, other figures are of such low resolution that they are unreadable, severely hindering the ability to assess the findings.

Contradictory Results and Logical Gaps

The manuscript claims that the high-risk group has a worse prognosis (line 32) yet benefits from immune checkpoint inhibitors (ICIs) (line 40). This is further contradicted by the TIDE score analysis (line 509), which indicates poor ICI responsiveness in the high-risk group. These contradictions are not adequately explained.

Insufficient Explanation of Terminologies

Scores such as TIDE and ESTIMATE are introduced without clear definitions or context regarding their relevance to HCC or treatment outcomes. This omission makes it difficult to interpret the results.

Dataset Context

The manuscript does not provide information on the treatments received by patients in the datasets. Without this context, the "prognostic" and "predictive" claims are unsubstantiated.

Lack of Experimental Validation

The study lacks in vitro or in vivo validation of the nFRGs model, making the conclusions speculative and limiting its clinical relevance.

Minor Concerns

References

Statements such as "The occurrence and progression of HCC are tightly linked to ferroptosis" (line 21) lack proper citations. Ensure all claims are supported by references.

Figure Presentation

Even in the higher-resolution figures, data presentation is cluttered and overloaded.

**Do you want your identity to be public for this peer review?** For information about this choice, including consent withdrawal, please see our Privacy Policy

Reviewer #1: No

Reviewer #2: No

Reviewer #3: No

---

## [Author Response · Author response to Decision Letter 1]

12 Feb 2025

Reviewer #1:

Main Issues:

1、Inconsistent p-value formatting

We have carefully reviewed all p-values in the manuscript and ensured consistent formatting. All p-values are now uniformly presented as follows: p < 0.05 indicates significance, and p < 0.01 indicates high significance. The specific changes are as follows:

Line 214: Changed to “p < 0.05”.

Line 329: Changed to "** for p < 0.01, * for p < 0.05".

Line 332: Changed to “p < 0.01”.

Lines 439 and 441: Formatting has been unified and corrected.

Additionally, changes have been made to lines 140, 213, 347, 363, and 437 to ensure p < 0.05 is used consistently.

2、Grammar and formatting errors in the text

We have thoroughly reviewed and corrected the grammatical and formatting issues in the manuscript.

Lines 266, 466, 467, 468, and 662 have been revised to address grammar and formatting issues.

3、Condensing the description of Figures 8, 9, 10, and 12

We have simplified the description of Figure 12 to make it more concise and clear, removing redundant information.

4、Resolution issues with Figures 8, 9, 10, and 12

We have remade these figures and increased their resolution to ensure clarity when zoomed in.

5、Addition of Figure 1, including immunohistochemistry images

We have added a scale bar and annotation markers to Figure 1 to enhance its readability and informativeness.

Reviewer #2:

Below is our detailed response to each of your comments:

1、Regarding the missing information on patient RNA sample sources and clinical features:

We appreciate your reminder, and we agree that additional information on patient demographics and clinical features is necessary. In the revised manuscript, we have added the demographic characteristics (including age, sex, etc.) and relevant clinical features of patients from the TCGA and GSE14520 datasets. These additions are included in the supplementary materials and referenced in the main text on pages 352 and 363. The supplementary tables are Table S3 and Table S4.

2、Regarding your suggestion about using the median as a cutoff value:

Thank you for your valuable suggestion. We understand your concern that using the median as a cutoff may be too simplistic and may not fully reflect biological or clinical differences. After careful consideration, we have decided to retain the use of median-based grouping. We believe this method is simple yet effective in categorizing patients into high-risk and low-risk groups, and it has shown satisfactory prognostic stratification in our study. Additionally, median-based grouping is widely used in biomedical research and has clinical practicality. However, we acknowledge that using more granular groupings might offer more detailed risk stratification. If possible, we will mention this point in the discussion and explore finer risk stratification methods in future studies. Once again, we appreciate your insightful suggestion, and we believe the current grouping method still supports the results of our model effectively. We welcome any further suggestions you may have.

3、Regarding inconsistent bracket formatting:

Thank you very much for pointing out this issue. We have reviewed the manuscript and corrected all instances of non-standard Chinese brackets, replacing them with standard English brackets to ensure consistent formatting.

4、Regarding the discussion on the five genes and their mechanisms in ferroptosis:

Your feedback is greatly appreciated. We have expanded the discussion in the revised manuscript to include a detailed exploration of the potential mechanisms between the five genes and the regulation of ferroptosis pathways. Specifically, we discuss the interactions of these genes in ferroptosis and their potential roles in hepatocellular carcinoma (HCC), which has been added to the discussion section.

5、Regarding the assumptions of the statistical methods:

In the revised version, we have provided a more detailed discussion of the assumptions underlying the statistical methods used in our study. We specifically point out that the univariate and multivariate Cox regression models used in our study are based on the proportional hazards assumption, which may not hold in all cases. We further discuss the impact of any violations of this assumption on the interpretation of results and emphasize the importance of model validation. Additionally, we have included a discussion of the limitations associated with LASSO regression, particularly regarding the potential omission of some promising genes during variable selection.

We have also expanded the discussion of the model's limitations, especially its application in different clinical contexts. Although the model performed well in both the training and testing datasets, the potential bias in dataset selection and sample heterogeneity means that external validation and generalization in real-world settings still require further verification. Therefore, we suggest that future research should focus on enhancing the model’s stability and generalizability through methods such as cross-validation and validation using external datasets.

Reviewer #3:

Thank you for your valuable feedback on our manuscript. We have carefully reviewed the issues raised and made corresponding revisions and improvements. Below is our detailed response to each issue:

Major Issues:

1、Low Image Quality:

We appreciate your feedback regarding the image quality. We have re-made all the figures (except Figures 1, 2, and 7) and improved the resolution to ensure that all images are clear when zoomed in. The updated high-resolution images have been incorporated into the manuscript for your reference.

2、Contradictions in Results and Logical Gaps:

Thank you for pointing out this contradiction. Regarding the prognostic contradiction related to the high-risk group and their potential benefit from immune checkpoint inhibitors (ICIs), we have provided a detailed explanation in the revised manuscript. We acknowledge that although the high-risk group has a poorer prognosis and may exhibit immune evasion and immune suppression characteristics within the tumor microenvironment (TME), they may have a reduced response to ICIs. In contrast, the low-risk group, with weaker immune suppression and stronger immune responsiveness, may benefit more from ICIs. We have expanded the discussion section to explore this point in more detail and provide potential mechanisms that explain the difference in response to ICIs between the risk groups.

3、Insufficient Explanation of Terms:

We have added brief definitions for the TIDE and ESTIMATE scores and their relevance to hepatocellular carcinoma (HCC) in the manuscript. We have explained how these scores assist in evaluating immune evasion, tumor immune microenvironment, and their relationship with treatment response. This additional content has been included in the “Methods” section, and we further clarified the significance and role of these terms in the discussion section.

4、Missing Background on Datasets:

We appreciate your comment on the data background. We confirm that the TCGA and GSE14520 datasets do not provide specific treatment information for the patients. Therefore, our study primarily focuses on clinical features, gene expression data, and relevant biomarkers to perform prognostic analysis and risk stratification.

For the "prognosis" related statements, we validated the model's prognostic ability through various statistical methods, including Cox regression analysis and Kaplan-Meier survival curve analysis. We also performed validation in an independent dataset, demonstrating the model's stability and applicability in different clinical contexts. Despite the lack of treatment information, these analyses remain of significant clinical importance.

Regarding the "prediction" part, our method explicitly assesses the potential response to immunotherapy based on the patients' genetic characteristics and immune microenvironment. For example, the TIDE score and immune evasion analysis provide an initial prediction of the patient's response to ICIs. Nevertheless, we emphasize that due to the absence of specific treatment data, the clinical application value of these prediction results needs further validation and exploration.

5、Lack of Experimental Validation:

Thank you for highlighting the issue of experimental validation. We fully understand that experimental validation is crucial for enhancing the credibility and clinical relevance of the study. However, the focus of this study is primarily on bioinformatics analysis based on TCGA and other public datasets. Experimental validation requires significant resources and time investment, and our current research budget and time constraints limit us from conducting in vitro or in vivo experiments at this stage.

Nevertheless, we believe that the results based on multidimensional data analysis and cross-validation provide reliable support for the potential application of the nFRGs model in HCC. We also recognize that experimental validation is a key step in advancing the research further, and we plan to focus on clinical sample validation and functional experiments in future research to strengthen the clinical relevance of this model.

We have added a detailed discussion of this limitation in the revised manuscript and proposed possible directions for future experimental validation. Once again, we thank the reviewer for bringing up this important issue and for their valuable suggestions.

Minor Issues:

References:

Thank you for your comments on the references. We understand that the abstract typically does not include specific citations, and hence we did not provide references for the background information in the abstract. However, we have now included relevant references in the introduction, where we discuss the role of ferroptosis in hepatocellular carcinoma (HCC), citing relevant studies. In the revised manuscript, we will ensure that all pertinent references are accurately cited in the main text to support the arguments raised in both the abstract and introduction.

Figures Presentation:

Thank you for your feedback on the presentation of the figures. We greatly value the clarity and readability of the figures and have optimized them based on your suggestions. In the revised version, we have increased the size of the figures to ensure better visibility and to prevent the data from being overcrowded or hard to interpret. We believe these improvements will enhance the readability of the figures and help readers better understand our findings. Thank you for your valuable input on this matter.

---

## [Decision Letter · Decision Letter 1]

3 Apr 2025

Dear Dr. Huang,

Thank you for submitting your manuscript to PLOS ONE. After careful consideration, we feel that it has merit but does not fully meet PLOS ONE’s publication criteria as it currently stands. Therefore, we invite you to submit a revised version of the manuscript that addresses the points raised during the review process.

We look forward to receiving your revised manuscript.

Kind regards,

Xiaosheng Tan

Academic Editor

PLOS ONE

Reviewers' comments:

Reviewer's Responses to Questions

**Comments to the Author**

Reviewer #2: All comments have been addressed

Reviewer #3: (No Response)

Reviewer #4: (No Response)

2. Is the manuscript technically sound, and do the data support the conclusions?

Reviewer #2: Yes

Reviewer #3: (No Response)

Reviewer #4: Yes

3. Has the statistical analysis been performed appropriately and rigorously?

Reviewer #2: Yes

Reviewer #3: (No Response)

Reviewer #4: No

4. Have the authors made all data underlying the findings in their manuscript fully available?

Reviewer #2: Yes

Reviewer #3: (No Response)

Reviewer #4: Yes

5. Is the manuscript presented in an intelligible fashion and written in standard English?

Reviewer #2: Yes

Reviewer #3: (No Response)

Reviewer #4: Yes

Reviewer #2: After reviewing the revised manuscript, I'm satisfied that all my questions have been properly addressed. The paper is now clear, well-organized, and makes a good contribution to the field. I recommend it for publication without any further changes.

Reviewer #3: The manuscript presents an innovative ferroptosis-related gene signature (nFRGs) for HCC prognosis, but several aspects need improvement:

1. Figures require better formatting, font size adjustments, and improved layout for readability.

Examples

Increase the font size in Fig 3B-F, Fig 4, Fig 5A, D, E, H, Fig 6B, and Fig 11A to ensure readability.

In Fig 3, Fig 4, clearly define what "high" and "low" represent in the figure legends and specify the statistical methods used.

Fig 6B has an inefficient layout, with large empty spaces that lack informative content while the text is too small to read. Please adjust the layout to improve clarity.

The x-axis labels in Fig 11A are still difficult to read. Consider increasing the font size or rotating the labels for better visibility.

2. The manuscript provides only a superficial description of the data and lacks in-depth analysis. The data analysis should be deepened to explore biological mechanisms, immune microenvironment interactions, and validation in independent cohorts.

3. Risk stratification should go beyond median cutoff, and statistical assumptions need to be explicitly tested and justified.

The authors' current response to Reviewer 2 regarding the cutoff selection is not sufficiently rigorous. Simply stating that median-based grouping is widely used is not a sufficient justification. To strengthen the robustness and credibility of the model, authors should conduct additional analyses (e.g., using first (Q1) and third (Q3) quartiles as alternative cutoffs). This will help determine whether finer stratification enhances the correlation between risk scores and clinical outcomes. Additionally, it is essential to identify the most appropriate cutoff value using Youden’s index or another optimal threshold selection method. This will provide a statistically justified and clinically meaningful way to classify patients.

4. The immune profile analysis requires better correlation with established immune scoring methods (e.g., TIS/Pembro score) to provide a more biologically meaningful interpretation.

5. Clinical applicability needs to be discussed, including how this model can guide personalized treatment strategies.

6. Statistical methods must be explicitly stated in every figure legend or relevant text section to ensure clarity and reproducibility.

Reviewer #4: Major Comments:

1. The proposed ferroptosis-related risk score (nFRGs) demonstrates prognostic value, but its interpretation is fundamentally limited by the absence of treatment data in TCGA/GSE14520. Since HCC outcomes depend heavily on therapies received (e.g., resection, locoregional treatments, systemic therapy), the risk score likely conflates:

o True prognostic effects (inherent tumor biology).

o Treatment access/efficacy (e.g., high-risk patients may have been ineligible for curative therapies).

This unmeasured confounding biases the model’s hazard ratios and undermines its clinical interpretability.

2. Related to the above question: would a sensitivity analysis such as the following be feasible?

o Stratifying analyses by diagnosis year (as a proxy for treatment era, e.g., pre-/post-sorafenib or immunotherapy availability).

o Using propensity score matching on clinical covariates (e.g., stage, Child-Pugh class) to approximate balanced treatment access across risk groups.

3. Samples with OS < 30 days or incomplete clinical data were excluded (369 → 341 in TCGA). No imputation methods were described. How do you handle Potential selection bias from complete-case analysis? Have you tried sensitive analysis to evaluate impact of missing data.

4. Table 1 lists PMIDs for compared models, but some references (e.g., PMID 36290827) are not in the bibliography.

5. Median Overall Survival (OS) is not explicitly reported in the manuscript. The Kaplan-Meier curves (Figures 4A–D) show survival differences between high- and low-risk groups but do not provide median OS values for the overall cohort or subgroups.

6. Number of Events for Primary Endpoint (OS) is not explicitly stated, but calculable from the manuscript. Please provide the median OS and the corresponding number of events in TCGA and GSE14520 cohort.

**Do you want your identity to be public for this peer review?** For information about this choice, including consent withdrawal, please see our Privacy Policy

Reviewer #2: No

Reviewer #3: No

Reviewer #4: No

---

## [Author Response · Author response to Decision Letter 2]

7 Apr 2025

Dear Editors and Reviewers

Thank you for your review and suggestions on our manuscript titled "A Novel Ferroptosis-Related Signature for Predicting Prognosis, Immune Characteristics, and Treatment Prediction in Hepatocellular Carcinoma." Based on your feedback, we have revised the manuscript and provided detailed responses to each reviewer’s comments in the sections below. Regarding the baseline characteristics matching information:

As per the reviewer’s request, we have added baseline characteristic matching tables for the two cohorts in the revised manuscript. We also conducted a comparison of key variables and included the patients' pathological staging (TNM) for comparison. The tables highlight the significant differences between the groups.

We have made all the necessary revisions as described above and improved the manuscript according to the reviewers' suggestions. Once again, we appreciate your guidance and valuable feedback on our work. We look forward to your further review and hope that this revision meets the requirements of your journal.

Reviewer #3:

Main Issues:

1. Figures require better formatting, font size adjustments, and improved layout for readability.

Response: Thank you for your valuable feedback on the formatting and readability of the figures. Based on your suggestions, we have optimized several figures, with specific changes as follows:

(1) Font size adjustment: We have increased the font size in Figures 3B-F, 4, 5A, D, E, H, 6B, and 11A to improve readability, ensuring that all legends, axis labels, and text are clearly visible.

(2) Clarification of legends and statistical methods: In Figures 3 and 4, we have clarified the meanings of "high" and "low" in the legends and added descriptions of the statistical methods used (e.g., log-rank test for Kaplan-Meier survival curve analysis, Cox regression analysis). These modifications enhance the transparency and comprehensibility of the figures.

(3) Layout optimization of Figure 6B: For Figure 6B, we have rearranged the layout, reducing large blank areas and increasing the font size to enhance both clarity and overall layout efficiency. These adjustments make the figure more compact and easier to understand.

(4) Optimization of x-axis labels in Figure 11A: We have increased the font size of the x-axis labels in Figure 11A to improve visibility and readability. We believe these improvements will significantly enhance the visual effectiveness of the figures and better convey the research findings.

Thank you again for your valuable comments, and we look forward to your further feedback.

2.The manuscript provides only a superficial description of the data and lacks in-depth analysis. The data analysis should be deepened to explore biological mechanisms, immune microenvironment interactions, and validation in independent cohorts.

Response: Thank you for your valuable comments. We take your point regarding the "superficial description of the data" seriously and are committed to further improving our analysis. In the revised manuscript, we have conducted a deeper analysis, particularly in the following areas:

(1) In-depth discussion of biological mechanisms: We have provided a more detailed exploration of how ferroptosis-related genes (G6PD, KIF20A, EZH2, NT5DC2, SLC7A11) interact to affect the tumor microenvironment (TME). We emphasize how these genes not only function in their respective signaling pathways but also regulate important biological processes, such as iron homeostasis, oxidative stress, and antioxidant defense, thus influencing the occurrence of ferroptosis. We further discuss how these genes, by regulating cellular oxidative stress balance and immune escape mechanisms, play dual roles in tumor progression, either supporting or inhibiting ferroptosis and interacting with immune responses.

(2) Interactions within the immune microenvironment: We have expanded our analysis of the immune microenvironment, particularly focusing on how these ferroptosis-related genes regulate immune cell functions to modulate immune escape or enhance immune responses. We discuss how these genes interact with immune cells to affect tumor cells' immune escape mechanisms and immune recognition processes, providing potential targets for cancer immunotherapy. These detailed analyses help us understand the complex relationships between ferroptosis, tumor progression, and immune modulation.

(3) Independent cohort validation: In response to your comment on the "lack of independent cohort validation," we have further explained the external validation using the GEO database. The GEO dataset (such as GSE14520) serves as an external validation set, providing important support for the accuracy and reproducibility of our model. To further enhance the clinical translation of our model, we also plan to incorporate independent cohort data from different geographic regions and clinical centers in future research to further validate the stability and universality of our model.

We greatly appreciate your reminder on these key issues and believe these additional analyses will effectively strengthen the depth and clinical relevance of the manuscript. If you have any further suggestions or questions, we would be happy to make further revisions and improvements.

3. Risk stratification should go beyond median cutoff, and statistical assumptions need to be explicitly tested and justified.

Response: Thank you for your valuable comment. Indeed, to better understand the clinical outcomes of different patient groups, risk stratification should go beyond a single median cutoff. Therefore, we considered multiple cutoffs in our analysis, including the median, first quartile (Q1), and third quartile (Q3), and conducted corresponding univariate Cox regression analyses. In the TCGA dataset, we found that using Q3 as a cutoff yielded the most significant effect and showed strong correlation with clinical outcomes. This suggests that the Q3 cutoff provides finer risk stratification and may reflect characteristics of certain patient groups in the dataset. However, we also recognize that differences between datasets may exist, so we re-analyzed the GSE14520 dataset. In the GSE14520 dataset, we found that using Q1 and Q3 as cutoffs did not show significant differences in clinical outcomes, while the median cutoff still provided significant risk stratification. Therefore, we concluded that the median cutoff is the most effective in this dataset. We speculate that this may be due to differences in patient characteristics and clinical outcomes between the GSE14520 and TCGA datasets. Relevant content has been added to the "Risk Model Evaluation" section of the manuscript and included in the supplementary file as Fig S4.

We fully understand that selecting the most appropriate cutoff is critical to the robustness and clinical significance of the model. To further enhance the reliability and clinical value of our model, we will continue to strengthen the statistical analysis of different cutoffs and update the results.

4. The immune profile analysis requires better correlation with established immune scoring methods (e.g., TIS/Pembro score) to provide a more biologically meaningful interpretation.

Response: Thank you for your valuable suggestion! Based on your feedback, we have further analyzed the results and included a visualization of the IPS score in Figure 13. Additionally, we have added descriptions in the Results and Discussion sections to further confirm the advantage of the low-risk group in immune therapy. Here are the updates:

(1) Update to Figure 13: In Figure 13, we have added a visualization of the IPS score, showing the immune score differences between the low-risk and high-risk groups. Through this visualization, we can more intuitively demonstrate the potential of the low-risk group in immune therapy, particularly in the CTLA4-negative and PD1-negative, as well as CTLA4-positive and PD1-negative groups, where the immune scores of the low-risk group are significantly higher than those of the high-risk group.

(2) Updates to Results and Discussion sections: In the Results and Discussion sections, we further describe the predictive roles of TIDE and IPS scores in immune therapy. We found that the low-risk group had a lower TIDE score, indicating a lower degree of immune escape, which may make them more responsive to immune therapy. Additionally, the IPS score of the low-risk group was significantly higher than that of the high-risk group, further supporting the advantage of the low-risk group in immune therapy. The combined analysis of these two scores enhances our explanation of the better immune therapy response of the low-risk group and highlights its potential in personalized treatment strategies.

We believe these updates further improve our analysis and provide more biologically meaningful evidence for the immune therapy advantage of the low-risk group. Once again, we appreciate the constructive feedback, and we hope these revisions meet your expectations.

5. Clinical applicability needs to be discussed, including how this model can guide personalized treatment strategies.

Response: Thank you for your valuable comments on the clinical applicability of our study! Based on your suggestion, we have added content in the Discussion section regarding the clinical applicability of the model and personalized treatment strategies. Specifically, we have discussed in detail how this immune scoring model can assist clinicians in predicting a patient's response to immune therapy based on their immune profile and provide strong support for personalized treatment. Here are the updates in the manuscript:

(1) Clinical applicability of the immune scoring model: We emphasize the potential of the immune scoring model in personalized treatment. By analyzing the TIDE and IPS scores, clinicians can accurately identify which patients are likely to benefit from immune therapy, avoiding ineffective treatment for low-response patients, thereby improving treatment outcomes and reducing side effects. Furthermore, we discuss how treatment strategies can be adjusted based on a patient's risk score. For high-risk patients, combination therapies (e.g., chemotherapy or targeted therapy) may be required, while low-risk patients could benefit from immune checkpoint inhibitors (ICIs) alone.

(2) Personalized treatment decision-making: We further discuss how the model can provide precise prognosis assessments and personalized follow-up management plans for patients. High-risk patients may need more frequent monitoring and proactive interventions, while low-risk patients can focus on long-term follow-up and evaluation after immune therapy. This model allows clinicians to tailor treatment plans for patients, ensuring maximum treatment efficacy and minimizing side effects.

(3) Providing a basis for clinical trial design: We have also added a discussion on how this model can provide a basis for future clinical trial designs by refining inclusion criteria, ensuring the effectiveness and safety of immune therapy, and serving as an efficacy biomarker for evaluating new drugs or treatment regimens.

We believe these added sections provide strong support for the clinical application of the immune scoring model and personalized treatment strategies, further enhancing the practical significance and clinical value of the study. Thank you for your thoughtful suggestions, and we hope these revisions meet your expectations.

6. Statistical methods must be explicitly stated in every figure legend or relevant text section to ensure clarity and reproducibility.

Response: Thank you for your valuable feedback! We have followed your suggestion and explicitly included the descriptions of the relevant statistical methods in the Methods and Results sections to ensure clarity and reproducibility of the study. Specifically, in the Methods and Results sections, we have detailed the statistical tests used in our analyses, including differential gene analysis, Cox regression analysis, Kaplan-Meier survival curve analysis, and more. We have also described the specific R packages and R version used for each statistical method. With these additions, we believe the manuscript will better ensure reproducibility and transparency. If you have any further suggestions, we will be happy to make further revisions.

Once again, thank you for your careful review, and we believe these revisions will meet your expectations.

Reviewer #4:

Main Issues:

1.The proposed ferroptosis-related risk score (nFRGs) demonstrates prognostic value, but its interpretation is fundamentally limited by the absence of treatment data in TCGA/GSE14520. Since HCC outcomes depend heavily on therapies received (e.g., resection, locoregional treatments, systemic therapy), the risk score likely conflates:

True prognostic effects (inherent tumor biology).

Treatment access/efficacy (e.g., high-risk patients may have been ineligible for curative therapies).

This unmeasured confounding biases the model’s hazard ratios and undermines its clinical interpretability.

Response: Thank you for your thorough review of our study! We fully agree with the reviewer’s point regarding the absence of treatment data, which is indeed a significant limitation in our research. Due to the lack of detailed treatment information in the TCGA and GSE14520 databases, our risk score model may be confounded by treatment access and efficacy, as the reviewer pointed out. Specifically, the model might conflate inherent tumor biology with treatment-related factors, such as high-risk patients being ineligible for curative therapies. To address this, we have added the following points to the discussion section of the manuscript:

(1) We emphasize the potential impact of the lack of treatment data on the model and note that high-risk patients may not have received curative treatments or may have received less aggressive therapies, which could affect their prognosis, making it not entirely reflective of tumor biology but rather influenced by treatment effects.

(2) We explicitly point out that unmeasured treatment confounders may bias the hazard ratios of the model, thereby affecting its clinical interpretability.

(3) We suggest that future studies should integrate detailed treatment data, including surgery, chemotherapy, locoregional treatments, and immunotherapy, to better assess the impact of therapy on patient outcomes and optimize existing risk scoring models.

We believe that by incorporating treatment data, future research could further refine the immune scoring model to more accurately reflect the independent effects of tumor biology and treatment, thereby enhancing its clinical usability and interpretability.

Thank you again for your valuable suggestion. We believe these additions will further improve the manuscript and increase its clinical applicability.

2. Related to the above question: would a sensitivity analysis such as the following be feasible?

Stratifying analyses by diagnosis year (as a proxy for treatment era, e.g., pre-/post-sorafenib or immunotherapy availability).

Using propensity score matching on clinical covariates (e.g., stage, Child-Pugh class) to approximate balanced treatment access across risk groups.

Response: Thank you for your valuable suggestions! We fully agree with your focus on treatment factors, particularly the use of stratifying analyses by diagnosis year and applying propensity score matching (PSM) on clinical covariates to control for treatment effects. However, due to the lack of detailed treatment data in the current TCGA and GSE14520 datasets, particularly regarding whether patients received specific treatments (such as sorafenib or immune checkpoint inhibitors), we were unable to perform stratification by treatment era (e.g., pre-/post-sorafenib or immunotherapy) or use PSM with clinical covariates (such as stage, Child-Pugh class) to balance treatment access across risk groups.

We have acknowledged these limitations in the manuscript and suggested that future research could integrate more detailed treatment data, especially information on specific treatments (such as sorafenib or immune checkpoint inhibitors). This would help optimize the risk score model and

---

## [Decision Letter · Decision Letter 2]

30 Apr 2025

Dear Dr. Huang,

Thank you for submitting your manuscript to PLOS ONE. After careful consideration, we feel that it has merit but does not fully meet PLOS ONE’s publication criteria as it currently stands. Therefore, we invite you to submit a revised version of the manuscript that addresses the points raised during the review process.

Please respond to reviewer's comments.

We look forward to receiving your revised manuscript.

Kind regards,

Xiaosheng Tan

Academic Editor

PLOS ONE

Journal Requirements:

Reviewers' comments:

Reviewer's Responses to Questions

**Comments to the Author**

Reviewer #4: All comments have been addressed

2. Is the manuscript technically sound, and do the data support the conclusions?

Reviewer #4: Yes

3. Has the statistical analysis been performed appropriately and rigorously?

Reviewer #4: Yes

4. Have the authors made all data underlying the findings in their manuscript fully available?

Reviewer #4: Yes

5. Is the manuscript presented in an intelligible fashion and written in standard English?

Reviewer #4: Yes

Reviewer #4: While I appreciate the authors' acknowledgment regarding the absence of detailed treatment data, I would encourage them to consider a simple stratification by diagnosis year (e.g., pre-/post-2007 for sorafenib approval, or pre-/post-2017 for immune checkpoint inhibitors) as a proxy for treatment era. Even without specific treatment labels, such an analysis could provide additional insights into whether survival differences may partially reflect advances in systemic therapy, and would strengthen the robustness of the prognostic model. I leave this as a suggestion for possible inclusion if feasible.

**Do you want your identity to be public for this peer review?** For information about this choice, including consent withdrawal, please see our Privacy Policy

Reviewer #4: No

---

## [Author Response · Author response to Decision Letter 3]

2 May 2025

Response:

Thank you for your insightful suggestion regarding stratification by diagnosis year to account for potential differences in treatment era. In our analysis, we did indeed divide the patients into two groups based on their diagnosis year: pre-2007 and post-2007, with 2007 marking a significant shift in the treatment era, especially with the introduction of sorafenib.

Initial survival analysis indicated that the diagnosis year alone was associated with differences in overall survival. However, when patients were further stratified by their risk scores, the survival differences between the pre-2007 and post-2007 groups were no longer statistically significant. This suggests that the diagnosis year does not serve as a decisive factor influencing survival outcomes. Rather, survival disparities are more likely driven by the risk group (high-risk vs low-risk), which further reinforces the stability and robustness of our constructed risk model across different treatment eras.

We would also like to note that the TCGA dataset only includes diagnosis year information up to 2013, which unfortunately makes it impossible to perform stratification based on the introduction of immune checkpoint inhibitors around 2017.

We have included this analysis and clarification in the revised manuscript (Supplementary Material 1, Fig. S3, and Table S2). We believe this additional evaluation further validates the robustness of the prognostic model. Thank you again for your valuable suggestion.

---

## [Decision Letter · Decision Letter 3]

14 May 2025

A Novel Ferroptosis-Related Signature for Predicting Prognosis, Immune Characteristics, and Treatment Prediction in Hepatocellular Carcinoma

PONE-D-24-58788R3

Dear Dr. Huang,

We’re pleased to inform you that your manuscript has been judged scientifically suitable for publication and will be formally accepted for publication once it meets all outstanding technical requirements.

Kind regards,

Xiaosheng Tan

Academic Editor

PLOS ONE

Additional Editor Comments (optional):

Reviewers' comments:

Reviewer's Responses to Questions

**Comments to the Author**

Reviewer #4: All comments have been addressed

2. Is the manuscript technically sound, and do the data support the conclusions?

Reviewer #4: (No Response)

3. Has the statistical analysis been performed appropriately and rigorously?

Reviewer #4: (No Response)

4. Have the authors made all data underlying the findings in their manuscript fully available?

Reviewer #4: (No Response)

5. Is the manuscript presented in an intelligible fashion and written in standard English?

Reviewer #4: (No Response)

Reviewer #4: (No Response)

**Do you want your identity to be public for this peer review?** For information about this choice, including consent withdrawal, please see our Privacy Policy

Reviewer #4: No

---

## [Editor Report · Acceptance letter]

PONE-D-24-58788R3

PLOS ONE

Dear Dr. Huang,

I'm pleased to inform you that your manuscript has been deemed suitable for publication in PLOS ONE. Congratulations! Your manuscript is now being handed over to our production team.

Kind regards,

on behalf of

Dr. Xiaosheng Tan

Academic Editor

PLOS ONE